# A DIFFUSION MODEL TO SHRINK PROTEINS WHILE MAINTAINING THEIR FUNCTION

**Ethan Baron**[*]
New York University

**Alan N. Amin**[*]
New York University

**Ruben Weitzman**
Harvard University

**Simon d'Oelsnitz**
Harvard University

**Debora S. Marks**
Harvard University

**Andrew G. Wilson**
New York University

## ABSTRACT

Many proteins useful in modern medicine or bioengineering are challenging to make in the lab, fuse with other proteins in cells, or deliver to tissues in the body because their sequences are too long. Shortening these sequences typically involves costly, time-consuming experimental campaigns. Ideally, we could instead use modern models of massive databases of sequences from nature to learn how to propose shrunken proteins that resemble sequences found in nature. Unfortunately, these models struggle to efficiently search the combinatorial space of all deletions, and are not trained with inductive biases to learn how to delete. To address this gap, we propose SCISOR, a novel discrete diffusion model that deletes letters from sequences to generate protein samples that resemble those found in nature. To do so, SCISOR trains a de-noiser to reverse a forward noising process that adds random insertions to natural sequences. As a generative model, SCISOR fits evolutionary sequence data competitively with previous large models. In evaluation, SCISOR achieves state-of-the-art predictions of the functional effects of deletions on ProteinGym. Finally, we use the SCISOR de-noiser to shrink long protein sequences, and show that its suggested deletions result in significantly more realistic proteins and more often preserve functional motifs than previous models of evolutionary sequences.

## 1 INTRODUCTION

As protein design becomes easier, more protein constructs are built for bioengineering, more protein medicines are being packaged for delivery to particular tissues, and, of course, more protein is being synthesized in the lab. Unfortunately, many important proteins are challenging to make, engineer, and deliver, due to their long sequences. Methods to build shorter versions of these proteins are expensive and often only narrowly applicable. Typically, experimentalists look for shorter homologues, which may not exist, and put them through costly optimization campaigns (Huang et al., 2022). Or, for proteins which function by well-characterized, simple biophysical interactions, experimentalists shrink sequences by running extensive physical simulations (Zhao et al., 2023).

Ideally we could instead learn how to shrink proteins using models trained on databases of protein sequences in nature – these models learn the constraints evolution has put on sequences across life and could shrink proteins to avoid breaking their function. Unfortunately, these large models (Notin et al., 2022; Nijkamp et al., 2022) struggle to effectively search through the massive space of all possible shrunken versions of a protein. They may also lack the inductive bias to predict the effect of deletions, having not been explicitly trained to do so. In principle, the first issue could be solved by diffusion models of protein sequences, like EvoDiff, which are effectively trained to plan series of many mutations and end with sequences that resemble those found in nature (Alamdari et al., 2023; Luo et al., 2022). However, current diffusion frameworks can only train models that perform substitution mutations – they cannot suggest deletions.

We propose a new diffusion model of evolutionary sequences that learns to generate by shortening sequences — Sequence Contraction with InSertion-Only noising pRocess (SCISOR). SCISOR adds noise to natural sequences by inserting random letters until they effectively become long random

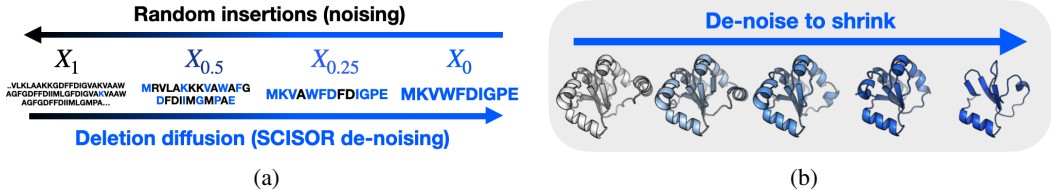

Figure 1: **SCISOR is a diffusion model trained to make deletions that arrive at a natural protein sequence. We can use it to shrink proteins while maintaining their function.** (a) We add random insertions to protein sequences from nature and train SCISOR to reverse these insertions. (b) Applying SCISOR diffusion to natural proteins, we get smaller proteins that are predicted to preserve parts of the tertiary structures of the original sequence. We show SCISOR samples of Q8NFU3 at 0, 5, 10, 20, and 50% deletion with structures predicted by OmegaFold (Wu et al., 2022).

sequences; then it trains a de-noiser to reverse this process by planning deletions that result in sequences that resemble those found in nature (Fig. 1a). Our contributions are:

- We introduce SCISOR, a **new discrete diffusion framework** that trains a de-noiser to generate sequences by learning to delete.
- We solve practical limitations of previous discrete diffusion models, enabling us to scale SCISOR up to large-scale evolutionary sequence data.
- We show that among large-scale diffusion models, SCISOR achieves **competitive model fit for protein sequences**.
- We show that the inductive biases of SCISOR allow it to make **state-of-the-art predictions of the effects of deletions on protein functions in the lab in ProteinGym**.
- Finally, we show that SCISOR **shortens proteins while better maintaining their structure and functional motifs** than methods using previous models of protein sequences.

## 2 BACKGROUND

Say we have a protein sequence $X$ made up of $L$ letters $X^{(1)}X^{(2)}\cdots X^{(L)}$ belonging to the alphabet of 20 amino acids $\mathcal{B}$. Our goal is to remove $M$ letters from $X$ to make a $\tilde{X} = X^{(j_1)}X^{(j_2)}\cdots X^{(j_{L-M})}$ with $j_1 < j_2 < \ldots, j_{L-M}$, that is still functional. Most random sets of deletions degrade the function of the protein, so we need to predict which deletions are unlikely to break the protein. Unfortunately there is very little data of sequence, shrunk-sequence pairs $(X, \tilde{X})$ to learn from; we must instead learn to predict functional shrunk proteins using other available data.

**Models of evolutionary sequences** One way we can learn how to shrink proteins is by learning from modern huge datasets of natural proteins. Indeed we can attempt to learn what a natural protein looks like in these databases; then we can pick a shrunken protein $\tilde{X}$ so that it looks natural and is therefore likely to be functional[1]. In practice, we can train huge generative models to generate natural proteins and use their likelihoods as a measure of naturalness (Riesselman et al., 2018; Rives et al., 2021; Notin et al., 2022; Nijkamp et al., 2022; Lin et al., 2023). Indeed, these likelihoods have been shown to be accurate predictors of whether single-letter-deletions will harm the function of a protein (Notin et al., 2022).

Unfortunately, the models that are typically used to fit this data, such as BERT-style (Rives et al., 2021; Lin et al., 2023) and autoregressive models (Notin et al., 2022; Nijkamp et al., 2022), struggle to search over the combinatorial space of all $\binom{L}{M}$ possible sets of deletions to find an ideal $\tilde{X}$. Ideally, we would have a model that can plan a number of deletions that arrive at a functional protein sequence. We also speculate that a model that learns directly how to delete would make more accurate predictions and designs.

**Discrete diffusion** To effectively search through a large mutational space, we could model the data with discrete diffusion models. These models generate samples by starting with a random sequence

---

[1]Note this does not guarantee our goal that $\tilde{X}$ have the same function as $X$. But if two functional proteins have similar sequences then they often have related function (Mistry et al., 2013). See also Sec. 9.

and applying mutations to arrive at a realistic sequence. In particular, a sequence is sampled from a simple distribution $X_1 \sim q(X_1)$ and then it is transformed from time $t = 1$ to $t = 0$ using a de-noiser $q_\theta((X_t)_{t=0}^1 \mid X_1)$ so that $X_0$ looks like a sequence from the data generating distribution (Campbell et al., 2022).

Diffusion models can therefore be used to search for sets of many mutations to a sequence, $X$, that result in a realistic looking sequence. To do so, one sets $X_s = X$ for some $s$ and then de-noises using the diffusion model by sampling a path $q((X_t)_{t=0}^s|X_s)$, giving a realistic $X_0$ near $X_s$. Indeed, this procedure has been used to suggest mutations to optimize sequences (Luo et al., 2022; Gruver et al., 2023).

To train a de-noiser $q_\theta$, we first define a forward process $p((X_t)_{t=0}^1)$ which takes samples from our target distribution $X_0 \sim p(X_0)$ and applies random noise to them from time $t = 0$ to $t = 1$, arriving at a distribution that is easy to approximate $p(X_1)$. Then we train the de-noiser to generate paths that match the paths of the forward process by optimizing an evidence lower bound (ELBO) as

$$\log q_\theta(X_0) \geq \mathbb{E}_{p((X_t)_{t=0}^1|X_0)} \log \frac{q_\theta((X_t)_{t=0}^1)}{p((X_t)_{t=0}^1|X_0)}. \tag{1}$$

Typically, however, the forward noising process is chosen to be random substitutions. Accordingly, the de-noiser $q_\theta$ only applies substitutions rather than deletions. To search over the space of deletions, we therefore need a new diffusion framework.

## 3 RELATED WORK

**Diffusion models with insertions and deletions**  In chemistry and language modeling, there have been diffusion models that have attempted to allow for insertions and deletions. Campbell et al. (2023) propose TDDM, a jump diffusion model to handle varying dimensionality, and Patel et al. (2025) train a TDDM model on language. Their forward noising process involves randomly deleting elements, such that the stationary distribution is an empty sequence. This allows them to train a model which can learn to expand sequences. As well, Johnson et al. (2021) formulate a discrete-time noising process for small-scale language modeling that includes insertions, deletions, and substitutions. Unfortunately their loss computation scales with the number of discrete time-steps and it is unclear how to extend their framework to continuous-time diffusion, which is known to be superior (Campbell et al., 2022). Contemporary with this work, like Johnson et al. (2021), Havasi et al. (2025) use auxiliary tokens to describe a flow-matching method for training a model to perform insertions and deletions in language.

Compared to these models, we train a diffusion model with inductive biases for the shrinking task. We also build models with competitive likelihoods to other generative models; in particular, unlike the models above (except Johnson et al. (2021)) our model has a closed-form ELBO, which allows principled model comparisons. We solve various mathematical and practical problems that ultimately allow us to do so:

- We prove that a formal stationary distribution is not necessary to define a diffusion model (Thm. 4.1). This allows us to train a diffusion model that only learns to shrink.
- We extend the derivation in Amin et al. (2025) to derive a "schedule conditioned" loss. This is more stable than classical discrete diffusion, and also allows us to condition on the number of deletions $M$ (Thm. 4.2).
- We Rao-Blackwellize our gradient estimator by integrating over all insertion paths. We do so in practice by noting a connection with sequence alignments (Prop. 4.3).
- We solve a number of engineering challenges to 1) pick an optimal rate function, 2) learn on very long sequences, and 3) leverage pretrained weights from ESM (Sec. 4.3).

**Leveraging evolutionary data to shrink proteins**  Recently, Raygun (Devkota et al., 2024) also suggested using a model trained on sequences from nature to shrink proteins. Raygun trains a stochastic autoencoder to embed and generate sequences of any length on the UniRef dataset which they apply to a variety of downstream tasks, including shrinking long proteins by decoding their embeddings at a shorter length. However, Raygun cannot enforce similarity between the shrunken sequence and original sequence. Furthermore, like previous generative models of protein sequences, Raygun was not specifically trained to shrink. Below we show that our model, SCISOR, suggests shrunken proteins that more often preserve structure and other indicators of function than Raygun.

## 4 A DIFFUSION MODEL THAT LEARNS TO DELETE: SCISOR

To search the space of deletions and train a model with the right inductive biases, in Sec. 4.1 we build a process which noises sequences by adding random insertions. Then in Sec. 4.2 we show how to train a de-noiser $q_\theta$ that reverses this process (Fig. 1a). Finally, in Sec. 4.3, we discuss the practical choices we made to efficiently train SCISOR. In Sec. 5 we describe how to use the de-noiser to generate sequences, shrink proteins, and plan deletions in practice.

### 4.1 FORWARD NOISING WITH THE PURE BIRTH PROCESS

We propose an insertion-only forward noising process for discrete diffusion known as the pure birth process (Kendall, 1948) with rate function $\beta(t)$ and insertion distribution $\pi$. Let $X_0$ be a sequence $X_0^{(1)}, \ldots, X_0^{(L)}$. There are $L+1$ possible locations we can insert letters. In the pure birth process, at instant $t$, each of these locations gains an insertion with rate $\beta(t)$. The letter that is inserted is drawn from some distribution $Y \sim \mathrm{Cat}(\pi)$. After $Y$ is inserted at some position, the process continues and there are now $L + 2$ positions in which there could be insertions with rate $\beta(t)$ (Fig. 1a). To train a diffusion model to reverse this process, we need to (1) easily sample $p(X_t|X_0)$ and (2) easily approximate $p(X_1|X_0)$.

**Sampling** $X_t$    Rather than simulate the pure birth process up until time $t$, we show in App. G.1 that $X_t$ can be sampled directly from $X_0$ as in Alg. 1. Note that $0 < \alpha(t) \leq 1$ controls how many insertions are added: by the property of negative binomial distributions, the expected length of $X_t$ is $\mathbb{E}[M_t + L] = \frac{L+1}{\alpha(t)} - 1$ which grows as $\alpha(t) \to 0$.

---

**Algorithm 1** Sample $X_t$

---

**Require:** Initial sequence $X = X^{(1)} \cdots X^{(L)}$, time $t$

1: Compute the probability of no insertions at a site $\alpha(t) \leftarrow \exp\left(-\int_0^t \beta(s)\, ds\right)$

2: Sample total number of insertions up to time $t$, $M_t \sim \mathrm{NegativeBinomial}(L+1, \alpha(t))$

3: Sample the number of insertions in each position by uniformly distributing $M_t$ into $L+1$ bins: $(\ell_0, \ldots, \ell_L) \sim \mathrm{UniformMultinomial}(M_t)$

4: **for** $j = 0$ to $L$ **do**

5:    Sample insertion $Y_j$ of length $\ell_j$, with each character independently from $\mathrm{Cat}(\pi)$

6: Add insertions into $X$ to construct $X_t \leftarrow Y_0 X^{(1)} Y_1 X^{(2)} \cdots X^{(L)} Y_L$

7: **return** $X_t$

---

**Approximating** $p(X_1|X_0)$    As $t$ grows, $X_t$ becomes longer. To build a diffusion model however, the distribution $p(X_t|X_0)$ typically must converge to a distribution so that it can be approximated by a distribution that can easily be sampled from, $q(X_1)$. Our critical insight is that $p(X_t|X_0)$, while not converging, can still be very well approximated by long random sequences as $t$ gets large.

**Theorem 4.1.** *(Proof in App. G.2) Say $X_0$ is a sequence with length $L$. Call $q(\cdot \mid L)$ a distribution over sequences of length $L$ which simply samples each letter independently from $\mathrm{Cat}(\pi)$. Then, as the number of insertions increases, $M_1 \to \infty$, $X_1$ becomes easier to approximate with $q$:*

$$\mathrm{KL}(p(X_1 \mid X_0, M_1)||q(X_1 \mid L + M_1)) \to 0. \tag{2}$$

### 4.2 LEARNING TO REVERSE THIS INSERTION-ONLY NOISING PROCESS

Given a forward process of insertions, we now wish to learn a de-noiser $q_\theta$ that generates sequences that resemble those found in nature by deleting letters from long random sequences. We now (1) describe our reverse process $q_\theta((X_t)_{t=1}^0)$, (2) write the ELBO in Eqn. 1 for our model, and (3) describe how the de-noiser $q_\theta$ is being trained toward a target that deletes letters that are unlikely to align with the starting sequence $X_0$.

**The reverse process**    For a forward path $(X_t)_{t=0}^1$ from a sequence $X_0$ of length $L$, define $t_1, \ldots, t_{M_1}$ to be the times of each insertion. We can then sample forward paths by first deciding how many insertions will occur until time 1 and when these insertions will occur, and then choosing what these insertions are:

$$p((X_t)_{t=0}^1|X_0) = p(M_1|L)p(t_1,\ldots,t_{M_1} \mid M_1, L) \prod_{M=1}^{M_1} p(X_{t_M}|X_{t_{M-1}}).$$

We follow the discrete diffusion framework in Amin et al. (2025) in defining the reverse process to match the noise schedule of the forward process. To generate a sequence of length $L$, we first decide the number of insertions and their times from the same distribution as $p$, and then denoise each insertion[2],

$$q_\theta((X_t)_{t=1}^0|L) = p(M_1|L)p(t_1,\ldots,t_{M_1} \mid M_1, L)q(X_1|L+M_1) \prod_{M=1}^{M_1} q_\theta(X_{t_{M-1}}|X_{t_M}, M).$$

Now we must only train our de-noiser $q_\theta(X_{t_{M-1}}|X_{t_M}, M)$ to take in a sequence $X_{t_M}$ and the number of insertions that sequence has $M$, and predict the sequence before the last insertion $X_{t_{M-1}}$. That is, $q_\theta(\cdot \mid X_{t_M}, M)$ can be thought of as a distribution over the letters of $X_t$.

**The loss** To train the de-noiser, we modify the calculation of the ELBO from Eqn. 1 as in Amin et al. (2025). We will then use this ELBO as our objective for training the de-noiser.

**Theorem 4.2.** *(Proof in App. G.3) Define $M_t$ as the number of mutations up to time $t$, and $\mathrm{prev}(X_t)$ is the last sequence that gained an insertion to become $X_t$. Then the negative log likelihood of a sequence of length $L$, $-\log q_\theta(X_0|L)$, is smaller than*

$$\mathbb{E}_{M_1}\mathrm{KL}(p(X_1 \mid X_0, M_1)||q(X_1|L+M_1))$$
$$+ \mathbb{E}_{t,X_t,M_t}\frac{M_t\beta(t)}{1-\alpha(t)}\mathrm{KL}(p(\mathrm{prev}(X_t) \mid X_0, X_t, M_t)||q_\theta(\mathrm{prev}(X_t) \mid X_t, M_t)) \quad (3)$$

The first term is the quantity in Eqn. 2 – how well we can approximate $p(X_1)$; it is small as long as $M_1$ is typically large, i.e. $\alpha(1)$ is small, and can be calculated as in App. B. The second term is the quantity we use to train the de-noiser. $q_\theta$ takes in $X_t$ and the number of insertions in $X_t$ and must predict which letter of $X_t$ was last inserted – $\mathrm{prev}(X_t)$. To train the model, we must calculate $p(\mathrm{prev}(X_t)|X_0, X_t, M_t)$.

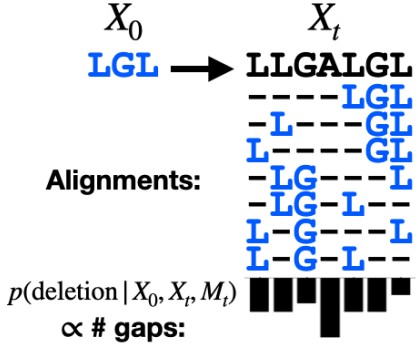

Figure 2: To calculate our target distribution of what letter to delete, $p(\mathrm{prev}(X_t) \mid X_0, X_t, M_t)$, we align our starting sequence $X_0$ to our noised sequence $X_t$. The reverse process should favor deleting letters that are gaps in more of the alignments.

**Target distribution** Eqn. 3 trains $q_\theta$ to match $p(\mathrm{prev}(X_t)|X_0, X_t, M_t)$, the true distribution over which letter of $X_t$ was last inserted in the forward process.

Conditioned on $X_0, X_t, M_t$, we could find $\mathrm{prev}(X_t)$ by simulating a pure birth process path from $X_0$ to $X_t$ and seeing what insertion occurred last. However there are multiple paths that could lead from $X_0$ to $X_t$; to calculate $p(\mathrm{prev}(X_t) \mid X_0, X_t, M_t)$, we must marginalize over all of these paths.

The next proposition shows that we can integrate over all of these paths by first enumerating every way to align $X_0$ to $X_t$ and noting that letters that align with $X_0$ less often are more likely to have been $\mathrm{prev}(X_t)$ (Fig. 2).

**Proposition 4.3.** *(Proof in App. G.4) Call $\mathrm{ali}(X, Y)$ the number of ways to align a sequence $X$ to a sequence $Y$. Call $b$ the letter that was deleted from $X_t$ to $\mathrm{prev}(X_t)$.*

$$p(\mathrm{prev}(X_t)|X_0, X_t, M_t) = \frac{\mathrm{ali}(X_0, \mathrm{prev}(X_t))}{M_t \cdot \mathrm{ali}(X_0, X_t)}.$$

Naively computing this quantity would require running an expensive alignment for every deletion. In practice, we use dynamic programming to compute all $\mathrm{ali}(X_0, \mathrm{prev}(X_t))$ in parallel (App. H).

## 4.3 IMPLEMENTING SCISOR AT UNIREF SCALE

Our ultimate goal is to train large SCISOR models on huge protein data – in particular, the UniRef database (Suzek et al., 2007). We train the SCISOR de-noiser with mini-batch gradient descent on

---

[2]Note our process is conditioned on generating a sequence of a particular length $L$.

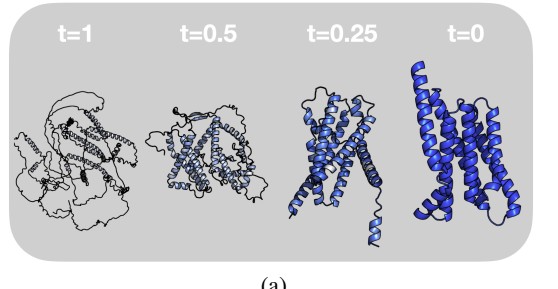 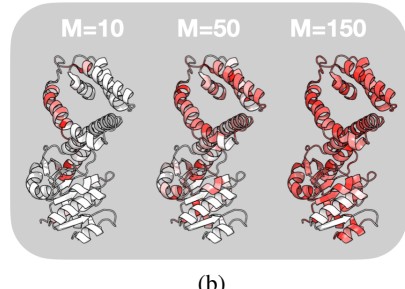

(a)                                         (b)

Figure 3: **The SCISOR de-noiser $q_\theta$ plans deletions to arrive at sequences that resemble those in nature, and therefore avoids deleting important structural motifs in natural sequences.** (a) SCISOR unconditionally samples proteins by starting with a large random sequence $X_1$ and iteratively deleting according to $q_\theta(\text{prev}(X)|X, M)$ to arrive at a protein that resembles those in nature. We predict the structure of each sequence with OmegaFold (Wu et al., 2022). (b) We ask SCISOR to plan the first of $M$ mutations for R4SNK4 and color residue $i$ on a structure from Aleku et al. (2016) by the deletion probability $q_\theta(X^{(-i)}|X, M)$. As $M$ increases, SCISOR favors deletions (red) in more regions while minimizing deletions in the catalytic structural motif near the bottom (white).

the second term of Eqn. 3 with i.i.d. samples of $t \sim \text{Uniform}(0, 1)$, $X_0, M_t, X_t$. We now discuss how we choose the rate function $\beta(t)$, the distribution of insertion letters $\pi$, the architecture for $q_\theta$, and methods to handle the large variation in sequence lengths of $X_t$ which we must pass to $q_\theta$.

**Hyperparameters** Our choice of hyperparameters follows that of standard diffusion methods. As in Austin et al. (2021); Amin et al. (2025), the rate function $\beta(t)$ was chosen so that the mutual information between $X_t$ and $X_0$ decreases roughly linearly on the interval $t \in [0, 1]$. We then modulated $\beta$ so that $\alpha(1)$ was large enough that the first term of Eqn. 3 is small, while samples in the second term did not get to many very long $X_t$. Details are in App. B. The categorical distribution $\pi$ was chosen to match the prevalence of amino acids in our training set.

**Architecture** We chose our architecture of the de-noiser $q_\theta(\cdot|X_t, M)$ to leverage the pre-trained weights of a BERT-style protein language model, while modifying the architecture to also condition on $M$. The ESM2 architectures (Lin et al., 2023) are trained on a masked language modeling task, taking in sequences and outputting logits at every site. We finetuned these models for $q_\theta$ by replacing their last layer with a linear and softmax layer. To condition on $M$, we add FiLM layers (Perez et al., 2017) between attention blocks, such that coordinate $d$ of the activations in layer $\ell$, $a_d^\ell$, is modified with an affine linear transformation $(1 + A_{\theta,d}^\ell(M)) \times a_d^\ell + B_{\theta,d}^\ell(M)$, where $A_\theta$ and $B_\theta$ are shallow fully-connected networks initialized to zero.

**Engineering for long sequences** Since $X_t$ sequences can have wildly different lengths, training naively could result in passing batches with a very high proportion of padding and passing very long sequences into the model. To avoid the first problem, we sort the $X_t$ sequences within a given batch by length, and pass them into the model in smaller sub-batches with accumulated gradients; this allowed us to reduce the proportion of compute spent on padding while maintaining an unbiased estimate of the loss. Next, to handle cases with extremely long $X_t$, if $|X_t| > 2048$, we randomly selected a window $X_t^{(w:w+2048)}$ uniformly at random to pass to the model. We then re-normalize the model predictions by $2048/|X_t|$ and use uniform predictions outside the window such that the deletion probabilities sum to 1. This choice keeps our ELBO a valid lower bound on the likelihood. Further details for how this impacts the ELBO and sampling are in App. B.

## 5 USING SCISOR TO GENERATE AND SHRINK PROTEIN SEQUENCES

The SCISOR de-noiser $q_\theta$ is trained as a generative model of natural sequences. In this section, we describe how to unconditionally generate natural sequences. Then we describe the statistical basis by which we may use the de-noiser for downstream tasks: unconditionally generating realistic protein sequences, predicting the effect of deletions on a protein's function, and shrinking long sequences to produce shorter natural sequences.

---

**Algorithm 2** Unconditional sequence generation with corrector steps

---

**Require:** Desired sequence length $L$, corrector steps $K$
 1: Sample $M \sim \text{NegativeBinomial}(L + 1, \, \alpha(1))$
 2: Sample $X$ of length $L + M$ where each $X^{(j)} \sim \text{Cat}(\pi)$ independently
 3: **while** $|X| > L$ **do**
 4:     **for** $k = 1$ **to** $K$ **do**
 5:         Remove one letter from $X$ according to $q_\theta(\text{prev}(X) \mid X, M)$
 6:         Insert a random letter from the distribution $\pi$ into a random position in $X$
 7:     Remove one letter from $X$ according to $q_\theta(\text{prev}(X) \mid X, M)$
 8:     $M \leftarrow M - 1$
 9: **return** $X$

---

**High-quality unconditional generation**  As described in Sec. 4.2, to sample a sequence of length $L$ from SCISOR, one samples a long random sequence from $\mathbb{E}_{M_1|L} q(X_1|L + M_1)$ and then iteratively deletes according to the de-noiser $q_\theta$ (Fig. 3a). Campbell et al. (2023) suggest continuous-time discrete diffusion models can get higher quality samples, sacrificing some compute, by applying corrector steps which noise and de-noise repeatedly. For SCISOR, this takes the form of adding and removing insertions as in Alg. 2. This allows SCISOR to more thoroughly search the space of deletions, potentially escaping local minima. In cases where doing many passes through the model is too expensive, we can make multiple deletions per de-noiser prediction, as discussed in App. B.

**Mutation effect prediction**  Say we have a sequence $X$ and we wish to predict the effect of the deletion of every position to understand the importance of each residue. Typically, we would take a model trained on protein sequences, $p_\theta$ and then evaluate the "natural-ness" of the sequence with each deletion $p_\theta(X^{(-i)})$ where $X^{(-i)}$ is the deletion of letter $i$ (Riesselman et al., 2018). Unfortunately estimating the likelihood is challenging for diffusion models as one needs to estimate the expectation in Eqn. 1.

SCISOR instead simply predicts $q_\theta(X^{(-i)} \mid X, M = 1)$ for every possible deletion $X^{(-i)}$. Then if the de-noiser suggests that a residue is unlikely to be deleted, that suggests that $X$ without that residue does not look like a sample from $q_\theta(X_0)$, i.e. a natural sequence, and thus that deletion may harm function. For multi-letter deletions, we integrate over all deletion paths (see App. B).

**Protein shrinking**  We now consider the problem of shrinking a sequence $X$ by $M$ deletions to a new sequence $\tilde{X}$ while preserving its function. One useful bias for this search may be using evolutionary information to look for $\tilde{X}$ which are substrings of $X$ which also look "natural". In practice, we can look for subsequences $\tilde{X}$ which are higher likelihood under a model trained on natural sequences. To do this, we can sample from $q_\theta(X_0 = \tilde{X} \mid X, M)$ which samples substrings in proportion to their "naturalness" $q_\theta(X_0 = \tilde{X})$. Indeed in App. D we show with three examples that deletions suggested by SCISOR correlate strongly with deletions seen in nature.

Note that SCISOR is not simply sampling deletions by how natural $\text{prev}(X)$ looks. Rather it also uses knowledge of $M$ to plan for future mutations. Different values of $M$ allow the model to change which deletions it will allow at each step (Fig. 3b). Given enough data and compute, SCISOR should learn the correct distribution $q_\theta(X_0 = \tilde{X} \mid X, M)$. In principle however, there is a distribution shift between sampling $q_\theta(X_0 = \tilde{X} \mid X, M)$ when $X$ is a realistic protein and our learning process when $X$ is a sequence with noisy insertions; this may impact the statistical efficiency of the learning process. In practice, App. D and our results in the following sections show that the SCISOR de-noiser learns meaningful evolutionary signals. We also confirm this using a toy example in App. I.

## 6  SCISOR IS A COMPETITIVE GENERATIVE MODEL FOR PROTEINS

We now compare how well SCISOR fits the distribution of natural sequences compared to established sequence modeling methods; we see SCISOR fits sequence data well, competitively with state-of-the-art diffusion and autoregressive models. All details are in App. C.

In Fig. 4, we compare the quality of SCISOR's fit to the data against state-of-the-art protein diffusion models, EvoDiff (Alamdari et al., 2023) and DPLM (Wang et al., 2024), and we include two autoregressive (AR) models from Alamdari et al. (2023) as references. All models are trained on

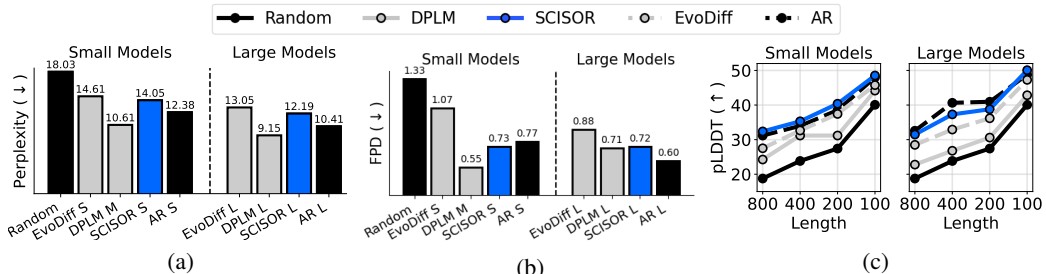

Figure 4: **SCISOR fits the distribution of sequences in nature competitively with established sequence modeling approaches.** (a) SCISOR is competitive with other diffusion models (grey) in perplexity. "S, M, L" refer to model size. (b, c) Samples from SCISOR ($K = 5$) are predicted to be competitive quality to those from diffusion models and competitive with AR models as measured by (b) matching the distribution of natural sequences as measured by the Fréchet protein distance (FPD) and (c) foldability (higher pLDDT from OmegaFold (Wu et al., 2022)). We took EvoDiff and AR perplexities from Alamdari et al. (2023).

the same release of UniRef50 (Suzek et al., 2007) – small models have 35-38M parameters, DPLM M has 150M parameters, and large models have 640-650M parameters. We evaluate each model's perplexity on a test set, and the quality of their samples, as measured by how well they match the distribution of natural sequences (FPD), and foldability (pLDDT).

Despite its difference from established modeling methods, SCISOR is competitive with other diffusion models in perplexities. As well, SCISOR often generates higher-quality samples than previous diffusion models, even competitive with the AR reference. As mentioned in Sec. 5, this is likely because SCISOR is a continuous-time model while the other diffusion models are discrete-time. In App. F.5, we ablate several components of our framework, particularly highlighting the importance of our Rao-Blackwellized training objective.

## 7 STATE-OF-THE-ART FUNCTIONAL EFFECT PREDICTIONS FOR DELETIONS

We evaluate SCISOR's ability to predict the effect of deletion mutations on the function of proteins as measured in the lab. To do so, we use 7000 measurements of deletion effects from 62 assays collected in ProteinGym (Notin et al., 2023). As baselines, we compare against existing mutation effect prediction models, including state-of-the-art autoregressive models ProGen2 (Nijkamp et al., 2022) and Tranception (Notin et al., 2022). Since models trained on UniRef90 tend to better predict the effects of mutations (Rives et al., 2021), all models in this section are trained on UniRef90.

In Fig. 5, we report the Spearman correlations of the measurements of each assay against the predicted effects from each model, taking the best performance within each model family (full table in App. F.1). SCISOR outperforms all baselines on both the single-deletion and multi-deletion benchmarks, even outperforming PoET (Truong & Bepler, 2023), a large model that has access to extra information about protein families.

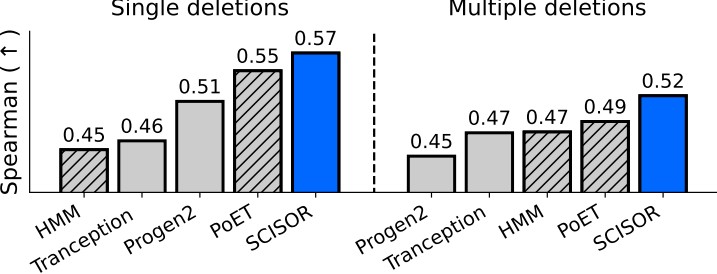

Figure 5: **SCISOR makes state-of-the-art predictions for the effect of deletions on protein function measured in the lab.** We calculate the average Spearman correlation between predicted deletion effects and measurements across all assays in ProteinGym, presenting the results from the highest-performing variant of each model architecture. Models that use multiple sequence alignment information are striped.

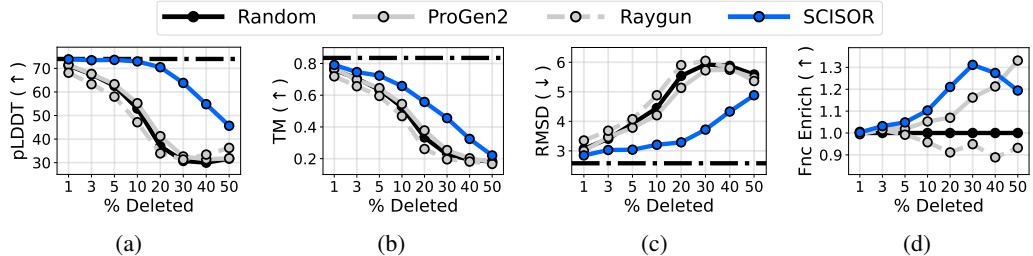

Figure 6: **SCISOR shrinks proteins while preserving their key properties.** We take 200 UniProt sequences with binding or active site annotations and shrink them by various amounts. We measure (a) pLDDT structural confidence scores from OmegaFold, predicted structural similarity between the original and shrunk proteins with (b) TM and (c) RMSD, and (d) conservation of annotated functional sites, measured by the enrichment ratio of annotated sites relative to random expectation. Horizontal dashed lines indicate reference values for 0% shrinking.

## 8 Preserving key in-silico indicators of function when shrinking

We now evaluate the ability of the SCISOR de-noiser $q_\theta$ to propose promising shrunk samples of long proteins from nature. Specifically, we take 200 sequences from UniProt that have binding or active site annotations and shrink them to various amounts.

Since the diffusion models in Sec. 6 cannot suggest deletions, we compare as baselines to shrinking with ProGen2 and Raygun. Raygun requires 1 model evaluation to make $M$ deletions, while SCISOR requires $M$. For ProGen2, ideally we would sample substrings of $X$ with length $L - M$ proportional to ProGen2's likelihood; however, that would require $\binom{L}{M}$ model evaluations, which is prohibitively expensive. We instead consider a strong but tractable baseline: we predict the effect of all $L$ single deletions independently (in $L$ model evaluations) and then sample sets of deletions with probabilities based on these effects. Further details are in App. C and App. F.3.

In Fig. 6 we see SCISOR consistently suggests shrunken proteins that are more likely to be foldable and preserve key indicators of function (predicted structural topology, presence of active or binding sites) than Raygun and ProGen2 baselines. Although ProGen2 achieves higher functional site preservation when shrinking by 50%, the lower pLDDT scores suggest that its samples at this level of shrinkage are not likely to be functional. We stratify these results by OmegaFold pLDDT, ESM2 pseudo-likelihood, and cellular localization in App. F.4 and App. F.6. In App. F.2, we show that SCISOR also achieves best performance for greedy sampling of shrunk sequences, reflecting situations in which a practitioner is not interested in generating diverse samples. Lastly, in App. E we perform an in-depth case study of shrinking the GTP sensor RalA, showing that shrinking with SCISOR best preserves the predicted structure of the binding site with GTP.

## 9 Conclusion

By proposing a new family of generative models that learn to build natural sequences by deleting, SCISOR, we have built models that can effectively shrink proteins. Future work may seek to address some of the conceptual limitations of the SCISOR process.

**Realistic insertion process** One way to mitigate the distribution shift mentioned in Sec. 5 is to make samples from $p(X_t)$ look more like natural sequences with a more elaborate forward process. The challenge is finding a process that provably achieves an easy-to-approximate $p(X_1)$ as in Sec. 4.1 and deriving a closed-form integral over all paths as in Sec. 4.2. Future work may leverage our theoretical and practical advances to derive the losses and train such models at large scale.

**Guiding based on function** In this work, we aimed to shrink proteins into sequences that may still appear in nature and are thus likely to be functional. While two functional proteins with similar sequences are likely to have the same function, this is not guaranteed, especially in those protein families with diverse functions (Zhang et al., 2024). Future work may incorporate other information of function into the SCISOR shrinking process. For example, one could guide the SCISOR diffusion process using a classifier trained to detect functional proteins of interest (Nisonoff et al., 2024).

**Including compensatory mutations** Currently, SCISOR only shrinks proteins via deletions. It is possible however that there are substitutions or insertions that could be added to a protein to make

it more tolerant to more deletions. To allow SCISOR to introduce these mutations while planning a series of deletions, we could add substitutions and deletions to the forward process, thereby training the de-noiser to also include substitutions and insertions in its planning.

ACKNOWLEDGMENTS

We thank Alex Ali and Andres Potapczynski for helpful feedback. This work was supported in part by NSF CAREER IIS-2145492, NSF CDS&E- MSS 2134216, NSF HDR-2118310, BigHat Biosciences, and Capital One.

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

## A  CODE RELEASE

We release our code at `https://github.com/baronet2/SCISOR` and model weights for SCISOR small (35M), medium (150M), and large (650M) trained on UniRef50 and UniRef90 at `https://huggingface.co/SCISOR/SCISOR`.

## B  DETAILS ABOUT SCISOR

### B.1  PRIOR MATCHING KL TERM

We rewrite the first term of Eqn. 3 so we can estimate it.

**Proposition B.1.** *(Proof in App. G)* $\mathrm{KL}(p(X_1 \mid X_0, M_1) \| q(X_1 | L + M_1))$ *is equal to*

$$\mathbb{E}_{X_1|X_0,M_1} \left[ \log \binom{M_1 + L}{L} + \sum_{i=1}^{L} \log \pi(X_0^{(i)}) - \log \mathrm{ali}(X_0, X_1) \right]. \tag{4}$$

We can therefore estimate the first term of the loss in Eqn. 3 by sampling $X_0, M_1, X_t$ and calculating the quantity in the expectation of Eqn. 4.

## B.2 EFFICIENT SAMPLING

Alg. 2 implements the Gillespie algorithm for a stochastic process. Zhao et al. (2024) and Amin et al. (2025) suggested $k$-Gillespie for diffusion models, taking $k$ steps at every step by sampling without replacement. Indeed we can do the same for SCISOR, sampling many deletions at each step without replacement.

## B.3 MULTI-DELETION PREDICTION

Say $\tilde{X}$ is the sequence $X$ with $M$ deletions at sites $\{i_1, \ldots, i_M\}$. We wish to calculate $q_\theta(X_0 = \tilde{X} \mid X, M)$. We can break this up into a sum over all deletions using the de-noiser

$$q_\theta(X_0 = \tilde{X} \mid X, M) = \sum_{m=1}^{M} q_\theta(X_0 = \tilde{X} \mid X^{(-i_m)}, M - 1) q_\theta(\mathrm{prev}(X) = X^{(-i_m)} \mid X, M).$$

Continuing like this, we can write $q_\theta(X_0 = \tilde{X} \mid X, M)$ as a sum over all permutations of the deletions.

---

**Algorithm 3** Predicting the functional effect of multiple deletions with SCISOR

---

**Require:** Initial sequence $X$, deletions $\{i_1, \ldots, i_M\}$.
1: $P \leftarrow$ all permutations of $\{i_1, \ldots, i_M\}$
2: SUM $\leftarrow 0$
3: **for** $j_1, \ldots, j_M \in P$ **do**
4:     SUM $=$ SUM $+ \prod_{M'=0}^{M-1} q_\theta(X^{(-j_1, \ldots, j_{M'+1})} \mid X^{(-j_1, \ldots, j_{M'})}, M')$
5: **return** SUM $= q_\theta(X_0 = \tilde{X} \mid X, M)$

---

## B.4 RATE FUNCTION

For simplicity, we choose a functional form

$$\beta(t) = \frac{\gamma}{1 - t_{\max} t}.$$

Consequently, we have:

$$\begin{aligned}
\alpha(t) &= \exp\left(-\int_0^t \beta(s)\, ds\right) \\
&= \exp\left(-\frac{\gamma}{t_{\max}} \int_0^t \frac{1}{1 - t_{\max} s}\, ds\right) \\
&= \exp\left(-\frac{\gamma}{t_{\max}} \ln(1 - t_{\max} t)\right) \\
&= (1 - t_{\max} t)^{\gamma/t_{\max}}
\end{aligned}$$

and $\alpha(1) = (1 - t_{\max})^{\gamma/t_{\max}}$. Now we must choose $\gamma$ and $t_{\max}$. We found empirically on small models that $\gamma = 1.1$ gave an ELBO (see Eq. 3) such that the expectation conditional on each $t$ was roughly even. We found empirically on small models that $t_{\max} = 0.9$ gave the best loss controlling for wall time, trading off allowing the model to attempt to fit larger sequences and spending compute on those large sequences.

### B.5 WINDOWING

One challenge in efficiently training the SCISOR de-noiser is that we must compute $q_\theta(\text{prev}(X_t) \mid X_t, M_t)$, where $X_t$ can potentially be a very long sequence. To handle these long sequences, we introduce a windowing strategy: if $|X_t| > 2048$, we randomly select a window $X_t^{(w:w+2048)}$ uniformly at random to pass to the model. We then re-normalize the model predictions by $2048/|X_t|$ (the probability of a deletion in the window is proportional to its size) and use uniform predictions outside the window such that the deletion probabilities sum to 1. Calling the predictions made by window $w$ $q_\theta^w(\text{prev}(X_t) \mid X_t, M_t)$, we can define our model predictions as an average over all windows

$$q_\theta(\text{prev}(X_t) \mid X_t, M_t) = \mathbb{E}_w q_\theta^w(\text{prev}(X_t) \mid X_t, M_t).$$

**ELBO** We modify the second term of our loss Eqn. 3 to obtain another lower bound to bring the expectation outside

$$\text{KL}(p(\text{prev}(X_t) \mid X_0, X_t, M_t) || q_\theta(\text{prev}(X_t) \mid X_t, M_t))$$
$$\geq \mathbb{E}_w \text{KL}(p(\text{prev}(X_t) \mid X_0, X_t, M_t) || q_\theta^w(\text{prev}(X_t) \mid X_t, M_t)).$$

This gives us a new ELBO we can estimate by stochastically sampling the window $w$ whenever we get a large sequence.

**Sampling** In Alg. 1, we need to sample from $q_\theta(\text{prev}(X_t) \mid X_t, M_t)$ for very long sequences. We do so by sampling a $w$ and then sampling from $q_\theta^w(\text{prev}(X_t) \mid X_t, M_t)$.

## C EXPERIMENTAL DETAILS

### C.1 BASELINES

We used **EvoDiff** models and code from `https://github.com/microsoft/evodiff` under the MIT license. We used **DPLM** models and code from `https://github.com/bytedance/dplm` under the Apache-2.0 license. We used **ProGen2** models and code from `https://github.com/enijkamp/progen2` under the BSD-3-clause license. We used **Raygun** models and code from `https://github.com/rohitsinghlab/raygun` under the CC BY-NC 4.0 license. We used **ProteinGym** models and code from `https://github.com/OATML-Markslab/ProteinGym` under the MIT license.

### C.2 SCISOR ARCHITECTURE

We used the flash attention implementation of ESM from Peng et al. (2024) under the MIT license. We used ESM2 weights (Lin et al., 2023) also under the MIT license. We developed SCISOR using code from `https://github.com/AlanNawzadAmin/SCUD` under the MIT license.

### C.3 TRAINING SCISOR

We apply our framework to train a protein generative model on UniRef50 (Suzek et al., 2007). We filter this dataset to exclude proteins with non-standard amino acids, and crop long protein sequences down to their first 1024 amino acids.

For the results in Section 6, we train SCISOR models on the March 2020 release of UniRef50, using the same train-test split as EvoDiff (Alamdari et al., 2023) from `https://zenodo.org/records/6564798`. Our models were trained about one week each on one NVIDIA A100 GPU with an effective batch size of 256 and learning rate of 0.0001.

For the results in Sections 7 and 8, we train SCISOR models on the latest release of UniRef90. Here, we use an effective batch size of 512 and learning rate of 0.00005. The SCISOR S and M models were trained for about one week each on two NVIDIA A100 GPUs. The SCISOR L model was trained for about four days on four NVIDIA H100 GPUs.

For each effective batch, we sampled all $t, X_0, M_t, X_t$. We then sorted sequences by the length of $X_t$ before breaking them into batches to pass to the model in batch sizes of 8 or 16; This makes sequences in each batch have similar length, minimizing padding.

## C.4 Model fit experiments

### C.4.1 Perplexities

**SCISOR** We compute the perplexity in Fig. 4a on the test dataset by first sub-sampling the expectation of the ELBO from Prop. 4.2 – we take 10 samples of $t, X_t$ for every sequence. We then divide by the total number of tokens in the test set and report the exponentiated negative result.

**EvoDiff and AR** We take perplexity values from Table S1 in Alamdari et al. (2023).

**DPLM** DPLM was trained as a discrete-time masking diffusion model with 500 steps and a linear rate schedule – that is, the probability of each token in $X_t$ being masked is $t/500$. We therefore evaluated their perplexities as such a model as in Austin et al. (2021). This ELBO becomes

$$\sum_{t=1}^{500} \frac{1}{t} \mathbb{E}_{X_0, X_t} \sum_{i=1}^{L} \mathbb{1}(X_t^{(i)} = \text{mask}) \log q_\theta(X_0^{(i)} \mid X_t).$$

### C.4.2 Samples

**SCISOR** We sampled according to Alg. 2.

**EvoDiff and AR** We sampled from EvoDiff and AR models using functions `generate_oaardm` and `generate_autoreg` from `https://github.com/microsoft/evodiff/blob/main/evodiff/generate.py`.

**DPLM** Wang et al. (2024) suggested a novel sampling method for DPLM. However, we were interested in measuring the quality of DPLM samples *as a diffusion model*. We therefore took samples as such a model as in Austin et al. (2021): We start with $X_{500}$ and for every $t = 500, \cdots, 1$ we unmask each position $i$ with probability $1/t$, replacing the mask according to predicted probabilities $q_\theta(X_0^{(i)} \mid X_t)$.

### C.4.3 Sample evaluation

For FPD we took 1000 protein lengths from UniRef50 and sampled sequences of each of those lengths from SCISOR, EvoDiff, and DPLM; or we sampled 1000 sequences from the AR models. For pLDDT, we sampled 100 sequences of length 100, 200, 400, and 800 from SCISOR, EvoDiff, and DPLM; for AR models where the sample length cannot be controlled, we sampled sequences until we had a sufficient number of samples with lengths within $10\%$ of each desired length.

**Fréchet protein distance (FPD)** We calculated the FPD of 1000 generated sequences to 10000 samples from UniRef50 using ProtT5 embeddings in `https://github.com/hefeda/PGP` under the Apache-2.0 license. We then calculated the Fréchet inception distance between the embeddings of the natural sequences and each set of sampled sequences as

$$\|\mu_{\text{natural}} - \mu_{\text{sample}}\|^2 + \text{tr}\left(\Sigma_{\text{natural}} + \Sigma_{\text{sample}} - 2(\Sigma_{\text{natural}}\Sigma_{\text{sample}})^{1/2}\right)$$

where $\mu$ and $\Sigma$ are empirical means and covariances of the embeddings.

**pLDDT** We calculate pLDDT scores using OmegaFold (Wu et al., 2022) as described in `https://github.com/HeliXonProtein/OmegaFold/blob/main/README.md` under the Apache-2.0 License. For computational efficiency, we use only 1 cycle per sample. This results in lower overall pLDDT scores than the recommended default settings, which uses 10 cycles to obtain more accurate predicted structures.

## C.5 ProteinGym

### C.5.1 Model predictions

**SCISOR** To evaluate SCISOR, we set $X_t$ to be the target sequence and $M$ to be the number of deletions between the target and the mutant of interest. We then predict the effect of the deletion using Alg. 3.

**ProGen and other models**  We evaluated other models using scripts available on ProteinGym.

### C.5.2  MODEL EVALUATION

For Fig. 5, we adapt the ProteinGym benchmark from (Notin et al., 2023) by filtering their indels dataset to cases where the mutant is a strict subsequence of the target sequence. For the single deletions benchmark, we use mutants that are only one deletion away from the target sequence, while for the multiple deletions benchmark, we use mutants that are two or three deletions away from the target sequence.

For single mutations, we gathered 61 assays in ProteinGym with 4544 mutations in total.

Three assays in ProteinGym measured double and triple mutations: `A4_HUMAN_Seuma_2022` measured stability and had 42 double mutations and 40 triple mutations, `KCNJ2_MOUSE_Macdonald_2022` measured expression and had 397 double mutations and 387 triple mutations, `P53_HUMAN_Kotler_2018` measured organismal fitness and had 172 double mutations and no triple mutations.

### C.6  SHRINKING

For Fig. 6a and 6d we sample 100 sequences with annotated active sites and 100 sequences with annotated binding sites from UniProt. We then shrink each sequence by $d$ percent, where $d \in \{1, 3, 5, 10, 20, 30, 40, 50\}$.

### C.6.1  MODEL SAMPLES

**SCISOR**  We shrunk sequences using Alg. 2.

**ProGen**  Ideally we could sample from $q_{\mathrm{ProGen}}(\tilde{X})$ over all shrunken versions of $X$, $\tilde{X}$, of desired length $L - M$. However, for even moderate values of $M$, this becomes computationally intractable. We therefore approximate this distribution by assuming each deletion has an independent effect:

$$\log q_{\mathrm{ProGen}}(\tilde{X}) \approx \log q_{\mathrm{ProGen}}(X) + \sum_{\text{deletions } i} \Delta_i$$

where $\Delta_i$ is the effect of a single mutation,

$$\Delta_i = \log \frac{q_{\mathrm{ProGen}}(X^{(i)})}{q_{\mathrm{ProGen}}(X)}.$$

This approximation requires calculating $L$ quantities $\Delta_i$.

Sampling from this approximation is equivalent to sampling $M$ deletions – deletion $i$ is sampled with probability proportional to $\exp(\Delta_i)$. Greedy shrinking just involves picking the $M$ mutations with the highest $\Delta_i$.

**Raygun**  We use the Raygun generate command to generate shrunken proteins of desired length, where length was calculated by first calculating rounded up number of deletions to introduce, and conditioning Raygun to generate a sequence of length $L - M$. We used a noise ratio of 0.5 with uniform sampling (noise sampled uniformly between 0 and 0.5), in order to limit the number of substitutions introduced. We use a filter ratio of 0.1 meaning we select the best candidate among ten generated sequences, and recycle sequences once.

### C.6.2  MODEL EVALUATION

We evaluate the foldability of the shrunk sequences using the average pLDDT per residue for the structure generated using OmegaFold (Wu et al., 2022) as described in `https://github.com/HeliXonProtein/OmegaFold/blob/main/README.md` under the Apache-2.0 License, using 1 cycle per sample. We calculate enrichment as the number of active or binding sites in the original sequence that were preserved in the shrunk sequence – we call a functional site "preserved" if no residues were modified or deleted.

## D   SCISOR DELETIONS MATCH THOSE IN NATURE

We collected sequences locally aligned to three proteins using the phmmer web server (Finn et al., 2011) (`open penalty=0.01`, `Extend=0.2`, and all other settings default) and counted how frequently each position of each sequence landed in a region that wasn't aligned ("Not aligned") or aligned with a gap token ("Gap"). These are two distinct ways to measure deletions made by nature. Next, we shrunk each sequence using SCISOR at various values of $M$, taking 512 samples for each $M$. We ran our analysis on a catalyst R4SNK4_9PSEU, a membrane transporter TPIS_HUMAN, and a transcription factor FOXA1_HUMAN.

In Fig. 7, we plot, for each position of each protein, how frequently we observe deletions as suggested by SCISOR versus as seen in nature. We see strong correlation between the samples from SCISOR and those from nature. Almost all comparisons are statistically significant – a random model never achieves a Spearman correlation above 0.015, while we achieve a p-value of $< 10^{-10}$ for each protein.

Interestingly, for R4SNK4_9PSEU we observe separate behavior at different $M$: we see qualitatively that at $M = 50$, SCISOR prefers making deletions in regions where gaps are observed in nature, while at $M = 100$ SCISOR prefers deleting large chunks of the protein, the region which was less frequently included in alignments.

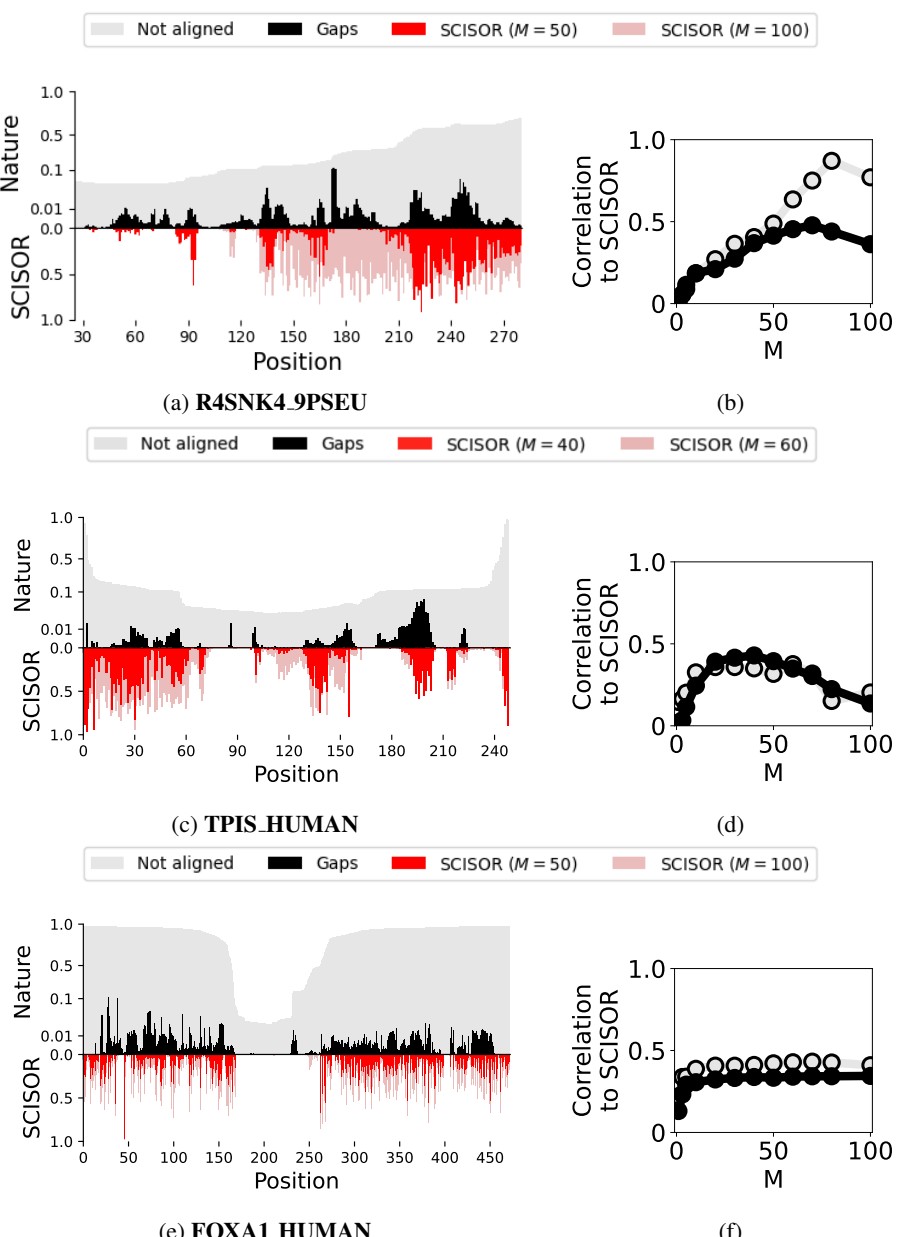

Figure 7: **SCISOR suggests deletions seen in nature.** (a, c, e) We compare two ways to measure deletions seen in an alignment versus deletions at two values of $M$ from samples of SCISOR. (b, d, f) We quantitatively measure the Spearman correlations between the two types of deletions in our alignment to deletions seen in SCISOR samples across $M$.

# E    CASE STUDY: SHRINKING RALA

To better understand SCISOR's ability to shrink natural proteins, we perform a detailed case study of the RalA protein, of length 206. We chose RalA since, as a GTP sensor, we could easily tell which sequences were unlikely to be functional by predicting their bound structure to GTP. We evaluate the function of shrunk samples by predicting their binding to GTP as in the structure `pdb_0000luad` and comparing to RalA. We use AlphaFold (Abramson et al., 2024) to predict the structure of shrunken RalA, GTP and binding partner Sec5.

**Qualitative results** We measure the TM score (a popular measure of structural similarity to ground truth (Zhang & Skolnick, 2004)) of wild type and shrunken RalA residues within 10 angstroms of the GTP molecule. Following Xu & Zhang (2010), we use TM=0.5 as a liberal cutoff for non-function and evaluate 5 designs for each method. The results are presented in Table 1. We see that proteins shrunk with SCISOR are much more likely to adopt the same complex fold as the wild type compared to designs from other methods.

| % Deleted | Random | Raygun | ProGen2 | SCISOR |
|---|---|---|---|---|
| 1 | 5 | 5 | 5 | 5 |
| 3 | 4 | 5 | 5 | 5 |
| 5 | 5 | 5 | 3 | 5 |
| 10 | 1 | 4 | 1 | 5 |
| 20 | 0 | 1 | 0 | 5 |

Table 1: Number of designs not predicted non-functional (TM scores above 0.5) (out of 5)

**Quantitative results** Next we show similar results with various measurements of the quality of the interface between the designed RalA and GTP. In Table 2, we perform a structural alignment using the RalA structures and then calculate the RMSD for all residues within 10 angstroms of the center of mass of the GTP. In Table 3, we instead calculate the RMSD of the atoms of the translated and rotated GTP. Additionally, we report DockQ values (Mirabello & Wallner, 2024) in Table 4 and the ipTM scores from AlphaFold in Table 5. In all these results, we see that SCISOR consistently outperforms all baselines, suggesting that the candidate designs from SCISOR are most likely to preserve the function of the original RalA protein.

| % Deleted | Random | ProGen2 | Raygun | SCISOR |
|---|---|---|---|---|
| 1 | $0.90 \pm 0.13$ | $0.87 \pm 0.10$ | $1.02 \pm 0.23$ | $0.92 \pm 0.01$ |
| 3 | $1.52 \pm 0.24$ | $1.25 \pm 0.15$ | $0.89 \pm 0.11$ | $0.91 \pm 0.01$ |
| 5 | $1.64 \pm 0.15$ | $1.48 \pm 0.37$ | $0.87 \pm 0.17$ | $0.93 \pm 0.00$ |
| 10 | $1.83 \pm 0.23$ | $1.93 \pm 0.28$ | $1.03 \pm 0.11$ | $1.24 \pm 0.14$ |
| 20 | $2.07 \pm 0.30$ | $2.28 \pm 0.18$ | $2.37 \pm 0.11$ | $\mathbf{1.28 \pm 0.15}$ |

Table 2: Interface RMSD of residues within 10 Å of GTP after structural alignment.

| % Deleted | Random | ProGen2 | Raygun | SCISOR |
|---|---|---|---|---|
| 1 | $1.69 \pm 0.30$ | $4.03 \pm 1.67$ | $7.39 \pm 4.82$ | $2.07 \pm 0.56$ |
| 3 | $3.91 \pm 2.04$ | $6.04 \pm 2.30$ | $3.47 \pm 0.53$ | $\mathbf{2.37 \pm 0.36}$ |
| 5 | $6.19 \pm 3.20$ | $3.53 \pm 0.18$ | $15.24 \pm 5.37$ | $3.50 \pm 0.72$ |
| 10 | $10.92 \pm 5.29$ | $11.47 \pm 6.11$ | $12.19 \pm 4.13$ | $\mathbf{3.95 \pm 1.24}$ |
| 20 | $17.83 \pm 5.55$ | $10.06 \pm 1.52$ | $15.24 \pm 1.89$ | $10.32 \pm 2.81$ |

Table 3: Ligand RMSD of GTP atoms after structural alignment.

| % Deleted | Random | ProGen2 | Raygun | SCISOR |
|---|---|---|---|---|
| 1 | $0.86 \pm 0.04$ | $0.71 \pm 0.17$ | $0.38 \pm 0.15$ | $0.87 \pm 0.01$ |
| 3 | $0.50 \pm 0.20$ | $0.51 \pm 0.18$ | $0.62 \pm 0.15$ | $\mathbf{0.85 \pm 0.02}$ |
| 5 | $0.46 \pm 0.18$ | $0.18 \pm 0.11$ | $0.23 \pm 0.13$ | $\mathbf{0.88 \pm 0.01}$ |
| 10 | $0.36 \pm 0.15$ | $0.18 \pm 0.15$ | $0.13 \pm 0.10$ | $0.50 \pm 0.18$ |
| 20 | $0.13 \pm 0.11$ | $0.11 \pm 0.07$ | $0.03 \pm 0.02$ | $0.30 \pm 0.16$ |

Table 4: DockQ scores for RalA binding to Sec5 with standard error.

| % Deleted | Random | ProGen2 | Raygun | SCISOR |
|---|---|---|---|---|
| 1 | $0.83 \pm 0.07$ | $0.82 \pm 0.08$ | $0.60 \pm 0.07$ | $\mathbf{0.90 \pm 0.00}$ |
| 3 | $0.64 \pm 0.11$ | $0.71 \pm 0.08$ | $0.82 \pm 0.08$ | $\mathbf{0.90 \pm 0.00}$ |
| 5 | $0.67 \pm 0.09$ | $0.52 \pm 0.08$ | $0.51 \pm 0.01$ | $\mathbf{0.89 \pm 0.00}$ |
| 10 | $0.40 \pm 0.14$ | $0.47 \pm 0.13$ | $\underline{0.59 \pm 0.07}$ | $0.67 \pm 0.09$ |
| 20 | $0.22 \pm 0.06$ | $0.36 \pm 0.06$ | $0.42 \pm 0.03$ | $\mathbf{0.58 \pm 0.09}$ |

Table 5: ipTM scores from AlphaFold for RalA binding Sec5.

# F    Supplementary Results

## F.1    Full ProteinGym Results

We show the results for ProteinGym for all models and sizes, stratifying the single deletions into functional, taxonomic, and MSA depth categories.

Table 6: ProteinGym results on single and multiple deletions.

| Model | MSA | Single Deletions | Multiple Deletions |
|---|---|---|---|
| ProGen2 S | | 0.457 | 0.445 |
| ProGen2 M | | 0.513 | 0.385 |
| ProGen2 Base | | 0.497 | 0.408 |
| ProGen2 L | | 0.491 | 0.375 |
| ProGen2 XL | | 0.393 | 0.392 |
| RITA S | | 0.409 | 0.274 |
| RITA M | | 0.448 | 0.318 |
| RITA L | | 0.465 | 0.323 |
| RITA XL | | 0.440 | 0.161 |
| Tranception S | | 0.439 | 0.475 |
| Tranception M | | 0.464 | 0.424 |
| Tranception L | | 0.445 | 0.426 |
| HMM | Yes | 0.453 | 0.474 |
| PoET (200M) | Yes | $\underline{0.551}$ | 0.488 |
| SCISOR S | | 0.332 | 0.268 |
| SCISOR M | | 0.505 | **0.520** |
| SCISOR L | | **0.573** | $\underline{0.492}$ |

Table 7: ProteinGym results on single deletions stratified by the measured function of each assay.

| Model | MSA | Activity | Expression | Organismal Fitness | Stability |
|-------|-----|----------|------------|--------------------|-----------|
| ProGen2 S | | 0.566 | 0.294 | 0.499 | 0.470 |
| ProGen2 M | | 0.574 | 0.404 | 0.558 | 0.514 |
| ProGen2 Base | | 0.592 | 0.380 | 0.496 | 0.520 |
| ProGen2 L | | 0.550 | 0.344 | 0.560 | 0.508 |
| ProGen2 XL | | 0.418 | 0.298 | 0.333 | 0.521 |
| RITA S | | 0.507 | 0.320 | 0.452 | 0.356 |
| RITA M | | 0.514 | 0.345 | 0.500 | 0.432 |
| RITA L | | 0.530 | 0.437 | 0.420 | 0.474 |
| RITA XL | | 0.532 | 0.385 | 0.360 | 0.481 |
| Tranception S | | 0.542 | 0.351 | 0.532 | 0.331 |
| Tranception M | | 0.594 | 0.340 | 0.526 | 0.395 |
| Tranception L | | 0.533 | 0.336 | 0.445 | 0.466 |
| HMM | Yes | 0.496 | 0.321 | 0.501 | 0.493 |
| PoET (200M) | Yes | **0.664** | **0.424** | 0.566 | 0.551 |
| SCISOR S | | 0.376 | 0.289 | 0.198 | 0.465 |
| SCISOR M | | 0.514 | 0.362 | 0.576 | 0.571 |
| SCISOR L | | 0.604 | 0.415 | **0.668** | **0.606** |

Table 8: ProteinGym results on single deletions stratified by the MSA depth of proteins in each assay.

| Model | MSA | Low | Medium | High |
|-------|-----|-----|--------|------|
| ProGen2 S | | 0.558 | 0.429 | 0.497 |
| ProGen2 M | | 0.415 | 0.483 | 0.544 |
| ProGen2 Base | | 0.438 | 0.460 | 0.568 |
| ProGen2 L | | 0.513 | 0.473 | 0.532 |
| ProGen2 XL | | 0.216 | 0.499 | 0.530 |
| RITA S | | 0.300 | 0.293 | 0.424 |
| RITA M | | 0.278 | 0.376 | 0.492 |
| RITA L | | 0.444 | 0.434 | 0.504 |
| RITA XL | | 0.139 | 0.462 | 0.508 |
| Tranception S | | 0.467 | 0.316 | 0.360 |
| Tranception M | | 0.297 | 0.358 | 0.447 |
| Tranception L | | 0.519 | 0.391 | 0.518 |
| HMM | Yes | 0.624 | 0.506 | 0.471 |
| PoET (200M) | Yes | 0.595 | 0.553 | 0.548 |
| SCISOR S | | 0.385 | 0.381 | 0.509 |
| SCISOR M | | **0.641** | 0.547 | 0.575 |
| SCISOR L | | 0.621 | **0.628** | **0.584** |

Table 9: ProteinGym results on single deletions stratified by the taxa of the protein in each assay.

| Model | MSA | Human | Eukaryote | Prokaryote | Virus |
|---|---|---|---|---|---|
| ProGen2 S | | 0.506 | 0.467 | 0.361 | 0.567 |
| ProGen2 M | | 0.536 | 0.539 | 0.432 | 0.510 |
| ProGen2 Base | | 0.568 | 0.541 | 0.396 | 0.471 |
| ProGen2 L | | 0.536 | 0.513 | 0.442 | 0.492 |
| ProGen2 XL | | 0.511 | 0.537 | 0.420 | 0.573 |
| RITA S | | 0.398 | 0.353 | 0.272 | 0.448 |
| RITA M | | 0.496 | 0.417 | 0.310 | 0.504 |
| RITA L | | 0.522 | 0.473 | 0.327 | 0.568 |
| RITA XL | | 0.510 | 0.466 | 0.386 | 0.555 |
| Tranception S | | 0.332 | 0.363 | 0.297 | 0.449 |
| Tranception M | | 0.443 | 0.406 | 0.295 | 0.468 |
| Tranception L | | 0.511 | 0.462 | 0.348 | 0.513 |
| HMM | Yes | 0.585 | 0.392 | 0.437 | 0.547 |
| PoET (200M) | Yes | 0.554 | 0.522 | 0.523 | 0.721 |
| SCISOR S | | 0.486 | 0.418 | 0.340 | 0.652 |
| SCISOR M | | 0.571 | 0.539 | 0.525 | 0.735 |
| SCISOR L | | **0.590** | **0.568** | **0.621** | **0.767** |

## F.2 DETERMINISTIC SHRINKING

In Section 8, we report results for sampling shrunk sequences from SCISOR and baseline models. Those evaluations are most relevant to practical settings, where practitioners may wish to generate a diverse set of promising candidate designs to be evaluated in the lab. Here, we instead consider deterministic shrinking, where the most likely deletions are performed. For Raygun, we mimic this setting by setting the noise parameter to 0. In Fig. 8, we see that SCISOR consistently achieves excellent performance in this setting as well. Although ProGen2 can achieve higher performance in some cases, this method is more computationally expensive than SCISOR. Moreover, our metrics may be rewarding the "myopic" nature of our ProGen2 baseline, which likely favors deletions that preserve functional regions in the original protein that are not necessarily functional in the shrunken protein.

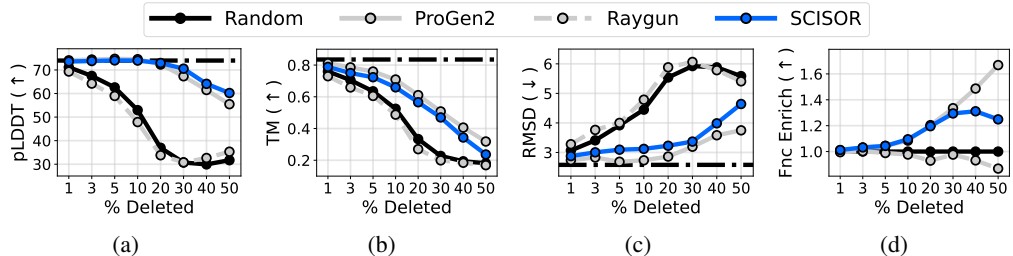

Figure 8: **Deterministically shrinking proteins with SCISOR yields high-quality shrunk samples.** All details are as in Fig. 6, except that we take the most likely deletions predicted from each model. For Raygun, we set noise $= 0$.

## F.3 COMPUTATIONALLY EXPENSIVE PROGEN2 BASELINE

In Section 8, we reported a baseline for ProGen2 that required $L$ forward passes of the model, already exceeding the $M$ forward passes used by SCISOR. That baseline assumes that the effects of each deletion are independent, which may be unrealistic. In Fig. 9, we present results for a more expensive ProGen2 baseline, requiring $\mathcal{O}(L \cdot M)$ forward passes, where deletions are performed one at a time, and the effects of subsequent deletions recalculated. We see that switching to this more expensive ProGen2 baseline does not improve performance, and SCISOR remains the best-performing method.

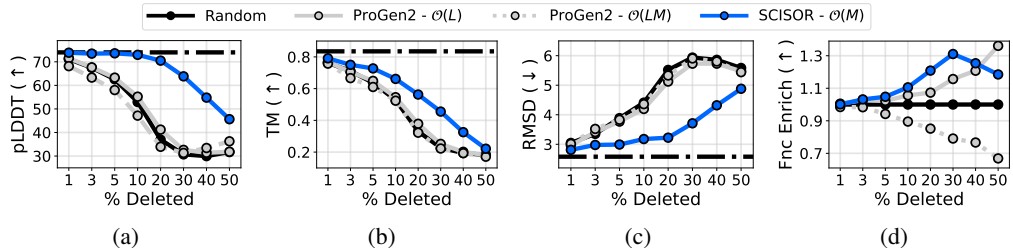

Figure 9: **SCISOR outperforms expensive ProGen2 baseline.** All details are as in Fig. 6, except that we include a more expensive ProGen2 baseline that requires $\mathcal{O}(L \cdot M)$ forward passes compared to SCISOR's $\mathcal{O}(M)$.

### F.4 SHRINKING RESULTS STRATIFIED BY WILD TYPE pLDDT AND ESM2 PSEUDO-LIKELIHOOD

In Fig. 10, we show how SCISOR's shrunk designs compare to the baselines, stratified by the OmegaFold pLDDT quartile of the wild type. SCISOR consistently produces high-quality designs across all settings, confirming its robustness.

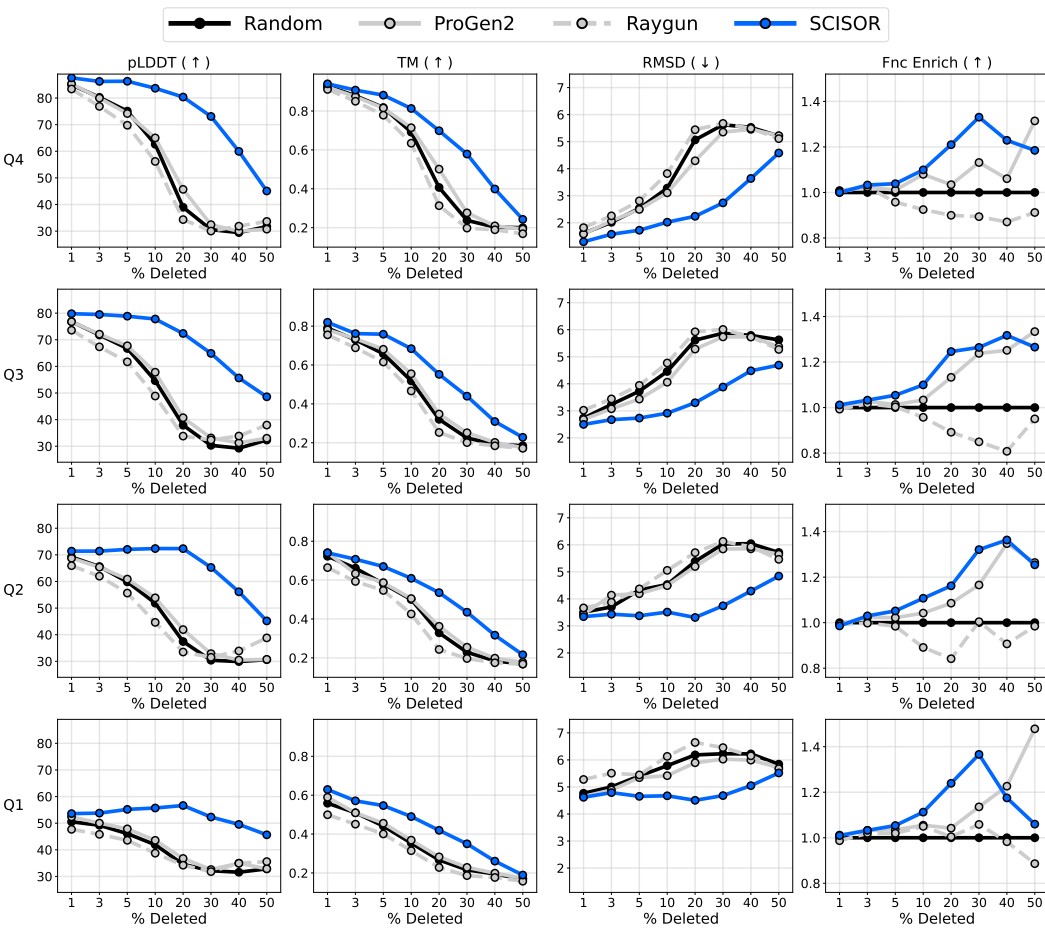

Figure 10: **SCISOR consistently predicts high-quality shrunk designs for wild type proteins with various pLDDT scores.** All details are as in Fig. 6, and we facet results by quartile of wild type pLDDT score, as predicted by OmegaFold.

In Fig. 11, we show how SCISOR's shrunk designs compare to the baselines, stratified by the ESM2 pseudo-likelihood quartile of the wild type, similar to the analysis in Gordon et al. (2024). The pseudo-likelihood is calculated by masking one index at a time, computing the likelihood of that element, and then aggregating the scores across the full sequence. SCISOR consistently produces high-quality designs across all settings, confirming its robustness.

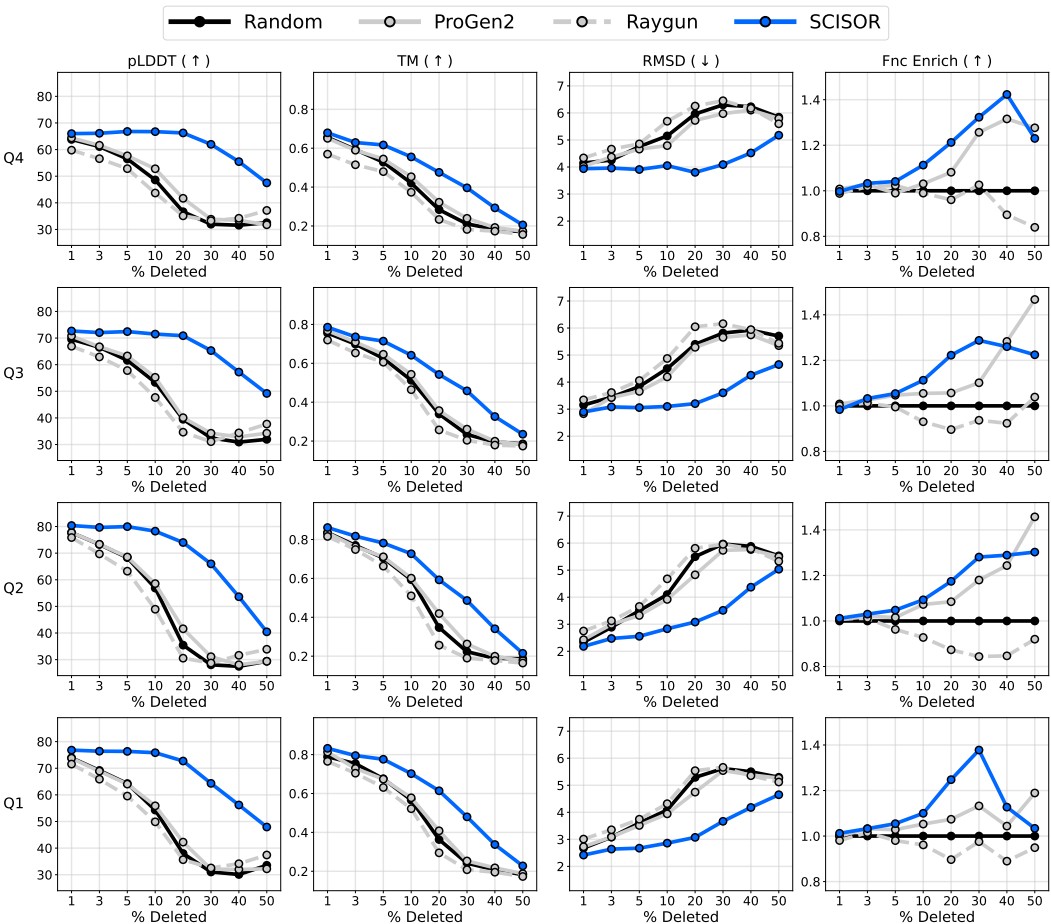

Figure 11: **SCISOR consistently predicts high-quality shrunk designs for wild type proteins with various pLDDT scores.** All details are as in Fig. 6, and we facet results by quartile of wild type ESM2 pseudo-likelihood.

## F.5 ABLATIONS OF MODELLING CHOICES

We perform an ablation study of training SCISOR S on UniRef50, ablating two aspects of our model. In Fig. 12a, we see that our structured forward process which weights amino acids by their prevalence results in a slight speedup of training. In contrast, in Fig. 12b, we see that the Rao-Blackwellization described in Section 4.2 is crucial in enabling the model to learn efficiently.

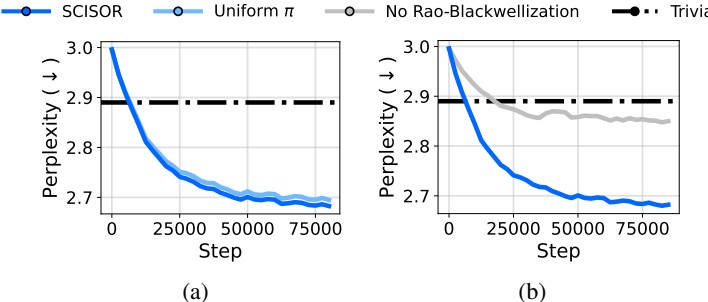

|   |   |
|:-:|:-:|
| (a) | (b) |

Figure 12: **Our structured forward process and Rao-Blackwellization of the loss enable efficient training of SCISOR.** We train SCISOR S on UniRef50 sequences, ablating components of our framework. While (a) switching from a uniform distribution over amino acids to one that weights amino acids by their prevalence results in a small improvement, (b) the Rao-Blackwellized loss is crucial to efficient training of SCISOR. The random baseline is based on the entropy of the amino acid distribution in UniRef50. We plot an exponential moving average of the perplexity on a withheld validation set.

### F.6 FEATURES ENRICHED IN PROTEINS THAT ARE HARD TO SHRINK

We investigated the differences between proteins in the analysis in Fig. 6, hypothesizing that membrane proteins may be more challenging to shrink. In Fig. 13 we find no enrichment in membrane proteins among the sequences whose structure dropped the most in quality after shrinking. This is potentially because they often contain large regions outside of the membrane, or because intra-membrane regions are not particularly sensitive to deletions. A preliminary analysis did reveal a slight enrichment of extra-cellular proteins however.

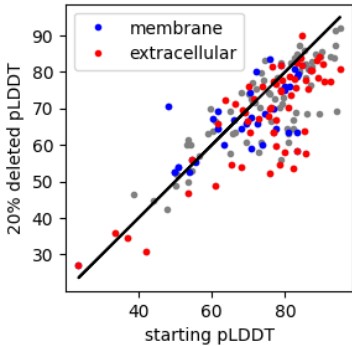

Figure 13: **We do not have evidence membrane proteins are harder to shrink.** We plot the results of Fig. 6 for 200 proteins. We plot the pLDDT of the full-length protein against that shrunk 20% by SCISOR. We colour membrane and extra-cellular proteins.

## G  PROOFS

### G.1  PROOF OF CORRECTNESS FOR ALGORITHM 1

The correctness of Alg. 1 follows from Cor. G.2.

**Theorem G.1.** *Call*

$$X_t = Y_0 X_0^{(1)} Y_1 X_0^{(2)} \cdots X_0^{(L)} Y_L$$

*where $X_0^{(1)} X_0^{(2)} \cdots X_0^{(L)}$ are the letters of $X_0$ and $Y_0, Y_1, \ldots, Y_L$ are the insertions. Then $|Y_l|$ is a* Geom$(\alpha(t))$ *distribution, where $\alpha(t) = \exp(-\int_0^t \beta(s)ds)$.*

*Proof.* By the Kolmogorov forward equation,

$$\frac{d}{dt}p(|Y_l| = n|t) = \beta(t)np(|Y_l| = n-1|t) - \beta(t)(n+1)p(|Y_l| = n|t).$$

This can be written as

$$\frac{d}{dt}\left(e^{(n+1)\int_0^t \beta(s)ds}p(|Y_l| = n|t)\right) = ne^{(n+1)\int_0^t \beta(s)ds}\beta(t)p(|Y_l| = n-1|t).$$

For $n = 0$, this is solved by $p(|Y_l| = 0|t) = \alpha(t)$. By induction,

$$p(|Y_l| = n|t) = \alpha(t)(1 - \alpha(t))^n$$

as

$$
\begin{aligned}
n\alpha(t)^{-(n+1)}\beta(t)p(|Y_l| = n-1|t) &= n\alpha(t)^{-n}\beta(t)(1-\alpha(t))^{n-1} \\
&= \frac{d}{dt}\left(\alpha(t)^{-n}(1-\alpha(t))^n\right)
\end{aligned}
\tag{5}
$$

$\square$

**Corollary G.2.**

$$p(|Y_0|, \ldots, |Y_L|) = \alpha(t)^{|X_0|+1}(1 - \alpha(t))^{\sum_l |Y_l|}$$

*so $p(|Y_0|, \ldots, |Y_L|)$ only depends on $|X_0|$ and $M = \sum_l |Y_l|$. In particular we can sample $M \sim$ NegativeBinomial$(\alpha(t))$ and then distribute it uniformly into $L + 1$ bins.*

*Proof.* Each $Y_i$ is generated independently, so we just take the product of probabilities from Prop. G.1. $\square$

### G.2 PROOF OF THEOREM 4.1

**Theorem G.3.** *(Proof of Thm. 4.1) Say $X_0$ is a sequence with length $L$. Call $q(\cdot \mid L)$ a distribution over sequences of length $L$ which simply samples each letter independently from $\mathrm{Cat}(\pi)$ for a distribution $\pi$ such that $\pi(b) > 0$ for all letters $b$. Then, as the number of insertions increases, $M_1 \to \infty$, $X_1$ becomes easier to approximate with $q$:*

$$\mathrm{KL}(p(X_1 \mid X_0, M_1)\|q(X_1 \mid L + M_1)) \to 0.$$

*Proof.* We suppress the subscript 1. Note by Lem. G.8

$$\frac{p(X \mid X_0, M)}{q(X \mid L + M)} = \frac{\mathrm{ali}(X_0, X)}{\binom{L+M}{L}\prod_{i=1}^L \pi(X_0^{(i)})}.$$

For a set of $L$ indices $I = i_1 < i_2 < \cdots < i_L$, call $\chi_I = \mathbb{1}(X_0 = X^{(i_1)}\cdots X^{(i_L)})$. Then $\mathrm{ali}(X_0, X) = \sum_I \chi_I$ and $E_q\chi_I = \prod_{i=1}^L \pi(X_0^{(i)})$. Therefore we can write

$$
\begin{aligned}
E_p \log \frac{p(X \mid X_0, M)}{q(X \mid L + M)} &= E_p \log \frac{\mathrm{ali}(X_0, X)}{\binom{L+M}{L}\prod_{i=1}^L \pi(X_0^{(i)})} \\
&= E_p \log \frac{\mathrm{ali}(X_0, X)}{E_q\mathrm{ali}(X_0, X)} \\
&\leq E_p \left| \frac{\mathrm{ali}(X_0, X)}{E_q\mathrm{ali}(X_0, X)} - 1 \right| \\
&= \frac{E_p |\mathrm{ali}(X_0, X) - E_q\mathrm{ali}(X_0, X)|}{E_q\mathrm{ali}(X_0, X)} \\
&\leq \frac{E_p |\mathrm{ali}(X_0, X) - E_p\mathrm{ali}(X_0, X)|}{E_q\mathrm{ali}(X_0, X)} \\
&\quad + \frac{|E_p\mathrm{ali}(X_0, X) - E_q\mathrm{ali}(X_0, X)|}{E_q\mathrm{ali}(X_0, X)} \\
&\leq \frac{\mathrm{Std}_p\left(\mathrm{ali}(X_0, X)\right)}{E_q\mathrm{ali}(X_0, X)} + \left| \frac{E_p\mathrm{ali}(X_0, X)}{E_q\mathrm{ali}(X_0, X)} - 1 \right|.
\end{aligned}
$$

We now show that these two terms each go to 0, starting with the second term.

**The second term** Say $X$ is generated by picking indices $Z = z_1 < \cdots < z_L$ which are $X_0$ and then generating all other letters from $\pi$ Say we have indices $I$. Then

$$
\begin{aligned}
E_p \chi_I &\leq (1 - p(I \cap Z = \emptyset)) + E_p \left[ \chi_I | I \cap Z = \emptyset \right] \\
&= 1 - \frac{\binom{M+L-L}{L}}{\binom{M+L}{L}} + E_q \chi_I \\
&\leq 1 - \left( \frac{M - L}{M} \right)^L + E_q \chi_I \\
&\leq O\left( L^2/M \right) + E_q \chi_I \\
&= (1 + o(1)) E_q \chi_I.
\end{aligned}
$$

Also

$$
\begin{aligned}
E_p \chi_I &\geq (1 - p(I \cap Z = \emptyset)) \times E_p \left[ \chi_I | I \cap Z = \emptyset \right] \\
&= \left( 1 - \frac{\binom{M+L-L}{L}}{\binom{M+L}{L}} \right) \times E_q \chi_I \\
&\geq \left( 1 - \left( \frac{M}{M + L} \right)^L \right) \times E_q \chi_I \\
&\geq \left( 1 - O\left( L^2/M \right) \right) E_q \chi_I \\
&= (1 - o(1)) E_q \chi_I.
\end{aligned}
$$

Then

$$
\frac{E_p \mathrm{ali}(X_0, X)}{E_q \mathrm{ali}(X_0, X)} = \frac{\binom{M+L}{L} E_p \chi_I}{\binom{M+L}{L} E_q \chi_I} = 1 + o(1).
$$

**The first term** We first change the expectation in the standard deviation into an expectation over $q$. Say $X$ is generated by picking indices $Z = z_1 < \cdots < z_L$ which are $X_0$ and then generating all other letters from $\pi$ Say we have indices $I, J$. Then

$$
\begin{aligned}
E_p \chi_I \chi_J &\leq (1 - p(I \cap Z, J \cap Z = \emptyset)) + E_p \left[ \chi_I \chi_J | I \cap Z, J \cap Z = \emptyset \right] \\
&\leq 1 - \frac{\binom{M+L-2 \times L}{L}}{\binom{M+L}{L}} + E_q \chi_I \chi_J \\
&= O(L^2/M) + E_q \chi_I \chi_J.
\end{aligned}
$$

We also have from above that

$$
E_p \chi_I E_p \chi_J = (1 + o(1)) E_q \chi_I E_q \chi_J.
$$

Then

$$
\mathrm{Var}_p \mathrm{ali}(X_0, X) = \sum_{I,J} \mathrm{Cov}_p(\chi_I, \chi_J) = \binom{M + L}{L}^2 o(1) + \sum_{I,J} \mathrm{Cov}_q(\chi_I, \chi_J).
$$

The first term is $o(1)$ against $E_q \mathrm{ali}(X_0, X)^2 = \binom{M+L}{L}^2 (E \chi_I)^2$, so we can just focus on the second term, $\mathrm{Var}_q \mathrm{ali}(X_0, X)$.

Note if $I \cap J = \emptyset$ then $\mathrm{Cov}_q(\chi_I, \chi_J) = 0$. Then

$$
\begin{aligned}
\sum_J \mathrm{Cov}_1(\chi_I \chi_J) &\leq \binom{M + L}{L} - \binom{M + L - L}{L} \\
&= o\left( \binom{M + L}{L} \right)
\end{aligned}
$$

using the same logic as above. Therefore, $\mathrm{Var}_q \mathrm{ali}(X_0, X) = o\left( \binom{M+L}{L}^2 \right) = o(E_q \mathrm{ali}(X_0, X)^2)$. This completes the proof.

$\square$

### G.3 PROOF OF THEOREM 4.2

**Theorem G.4.** *(Proof of Thm. 4.2) Define $M_t$ as the number of mutations up to time t, and $\mathrm{prev}(X_t)$ is the last sequence that gained an insertion to become $X_t$. Then the negative log likelihood of a sequence of length L, $-\log q_\theta(X_0|L)$, is smaller than*

$$\mathbb{E}_{M_t}\mathrm{KL}(p(X_1 \mid X_0, M_1)\|q(X_1|L + M_1))$$
$$+ \mathbb{E}_{t,X_t,M_t}\frac{M_t\beta(t)}{1-\alpha(t)}\mathrm{KL}(p(\mathrm{prev}(X_t) \mid X_0, X_t, M_t)\|q_\theta(\mathrm{prev}(X_t) \mid X_t, M_t))$$

*Proof.* The proof of Prop. 4.4 from Amin et al. (2025) derives an ELBO

$$\mathbb{E}_{M_t}\mathrm{KL}(p(X_1 \mid X_0, M_1)\|q(X_1|L + M_1))$$
$$+ \mathbb{E}_{t,X_t,M_t}w(M_t,t,X_0)\mathrm{KL}(p(\mathrm{prev}(X_t) \mid X_0, X_t, M_t)\|q_\theta(\mathrm{prev}(X_t) \mid X_t, M_t))$$

where

$$w(M_t, t, X_0) = \lim_{\epsilon\to 0} E[\#\text{events in } [t-\epsilon, t]|M_t, X_0]/\epsilon.$$

The following lemma shows this result. $\qquad\square$

**Lemma G.5.**

$$w(M, t, X_0) = M\frac{\beta(t)}{1-\alpha(t)}$$

*Proof.* First we change our time variable to $\tau = -\log\alpha(t)$. Noting $-\log\alpha(t-\epsilon) = \tau - \epsilon\beta(\tau) + O(\epsilon^2)$, we have

$$w(M, t, X_0) = \beta(t)\lim_{\epsilon\to 0} \tilde{\mathbb{E}}[\#\text{events in } [\tau-\epsilon, \tau]|M_\tau = M]/\epsilon \qquad (6)$$

where $\tilde{\mathbb{E}}$ is as if the rate $\beta$ were constant. In SCUD, events occur uniformly in the time interval, so the RHS would be $M/\tau = M/(-\log\alpha(t))$. For SCISOR, events are more concentrated later in time since more insertions increases the rate of insertion.

$$\tilde{\mathbb{E}}[\#\text{events in } [\tau-\epsilon, \tau]|M_\tau = M] = P[M_{\tau-\epsilon} = M - 1|M_\tau = M] + O(\epsilon^2)$$
$$= \frac{P[M_{\tau-\epsilon} = M - 1]}{P[M_\tau = M]}P[M_\tau = M|M_{\tau-\epsilon} = M - 1] + O(\epsilon^2). \qquad (7)$$

Note

$$P[M_\tau = M \mid M_{\tau-\epsilon} = M - 1] = P[M_\tau \geq M \mid M_{\tau-\epsilon} = M - 1] + O(\epsilon^2)$$
$$= P(\mathrm{Exp}(M + |X_0|) \leq \epsilon) + O(\epsilon^2)$$
$$= 1 - e^{-\epsilon(M+|X_0|)} + O(\epsilon^2) \qquad (8)$$
$$= \epsilon(M + |X_0|) + O(\epsilon^2).$$

Finally,

$$\frac{P[M_\tau = M - 1]}{P[M_\tau = M]} = \frac{\mathrm{NegBin}(|X_0|, e^{-\tau}; M - 1)}{\mathrm{NegBin}(|X_0|, e^{-\tau}; M)}$$
$$= \frac{\binom{m-1+|X_0|}{m-1}(1-e^{-\tau})^{M-1}}{\binom{M+|X_0|}{M}(1-e^{-\tau})^M} \qquad (9)$$
$$= \frac{M}{(M + |X_0|)(1-e^{-\tau})}.$$

This gives us

$$w(M, t, X_0) = M\frac{\beta(t)}{1-\alpha(t)}$$

which is similar for small alpha to the SCUD weight of $w(M, t, X_0) = M\frac{\beta(t)}{-\log\alpha(t)}$ but becomes larger at larger values. $\qquad\square$

### G.4 Proof of Proposition 4.3

**Proposition G.6.** *(Proof of Prop. 4.3) Call* $\mathrm{ali}(X, Y)$ *the number of ways to align a sequence* $X$ *to a sequence* $Y$.

$$p(\mathrm{prev}(X_t)|X_0, X_t, M_t) = \frac{\mathrm{ali}(X_0, \mathrm{prev}(X_t))}{M_t \cdot \mathrm{ali}(X_0, X_t)}.$$

*Proof.* Say $Y_t$ is $X_t$ with a single deletion, the letter $b$. By Lem. G.8

$$
\begin{aligned}
p(Y_t \mid X_0, X_t, M_t) &= \frac{p(Y_t \mid X_0, M_t - 1)}{p(X_t \mid X_0, M_t)} p(X_t \mid Y_t) \\
&= \frac{\binom{L+M_t-1}{L}^{-1} \mathrm{ali}(X_0, Y_t)}{\binom{L+M_t}{L}^{-1} \mathrm{ali}(X_0, X_t) \pi(b)} \pi(b)(L + M_t)^{-1} \\
&= \frac{(L + M_t) \mathrm{ali}(X_0, Y_t)}{M_t \mathrm{ali}(X_0, X_t)} \\
&= \frac{\mathrm{ali}(X_0, Y_t)}{M_t \mathrm{ali}(X_0, X_t)}.
\end{aligned}
$$

Note finally that we've ignored that there may be multiple deletions that take $X_t$ to $Y_t$ when calculating $p(X_t \mid Y_t)$. We can safely do so as it does not affect the loss Eqn. 3 of any of our other algorithms. $\square$

### G.5 Derivation of prior matching KL term

**Proposition G.7.** *(Proof of Prop. B.1)* $\mathrm{KL}(p(X_1 \mid X_0, M_1) || q(X_1 | L + M_1))$ *is equal to*

$$\mathbb{E}_{X_1 | X_0, M_1} \left[ \log \binom{M_1 + L}{L} + \sum_{i=1}^{L} \log \pi(X_0^{(i)}) - \log \mathrm{ali}(X_0, X_1) \right].$$

*Proof.* From Lem. G.8,

$$p(X_1 \mid X_0, M_1) = \binom{M_1 + L}{L}^{-1} \mathrm{ali}(X_0, X_1) \prod_{b \in X_1 \setminus X_0} \pi(b).$$

Given that $q(X_1 | L + M_1) = \prod_{b \in X_1} \pi(b)$, this finishes the proof. $\square$

### G.6 Useful lemma

**Lemma G.8.** *Calling the letters in* $X_t$ *that are not in* $X_0$ $X_t \setminus X_0$,

$$p(X_t \mid X_0, M_t) = \binom{L + M_t}{L}^{-1} \mathrm{ali}(X_0, X_t) \prod_{b \in X_t \setminus X_0} \pi(b)$$

*Proof.* To generate $X_t$ from $X_0, M_t$, we could (1) decide which positions $i_1, \dots, i_L \in 1, \dots, L + M_t$ should come from $X_0$ and then generate the rest of the letters according to $\pi$. Then

$$
\begin{aligned}
p(X_t \mid X_0, M_t) &= \sum_{\text{indices } i_1, \dots, i_L} \frac{\mathbb{1}(X_0 = X_t^{(i_1)} \cdots X_1^{(i_l)})}{\binom{L+M_t}{L}} \times \prod_{b \in X_t \setminus X_0} \pi(b) \\
&= \binom{L + M_t}{L}^{-1} \mathrm{ali}(X_0, X_t) \prod_{b \in X_t \setminus X_0} \pi(b).
\end{aligned}
$$

$\square$

## H  ALIGNMENTS ALGORITHM

Both KL terms in the ELBO make use of the primitive $\mathrm{ali}(X, Y)$. In particular, the denoising KL term requires computing the number of alignments between $X_0$ and each possible $\mathrm{prev}(X_t)$, a total of $|X_t|$ computations. Naively, computing the alignments between each pair of sequences takes $\mathcal{O}(|X_0| \cdot |X_t|)$ time for a total of $\mathcal{O}(|X_0| \cdot |X_t|^2)$. However, we devise an efficient dynamic programming algorithm to compute all of the alignment terms in parallel in $\mathcal{O}(|X_0| \cdot |X_t|)$ time, presented in Algorithm 4.

---

**Algorithm 4** Compute $\mathrm{ali}(X_t^{-(l)}, X_0)$ for all $l$ in parallel

---

**Require:** Sequences $X_0$ with $|X_0| = L$ and $X_t$ with $|X_t| = N$
1: Set $\mathrm{matching}[i, j] \leftarrow \mathbb{I}(X_0^{(i)} = X_t^{(j)})$ for all $i \in \{1, \ldots, L\}, j \in \{1, \ldots, N\}$
2: Initialize $\mathrm{prefix\_dp} \leftarrow \mathbf{0}^{(N+1) \times (L+1)}$
3: Set $\mathrm{prefix\_dp}[i, 0] \leftarrow 1$ for all $i \in \{1, \ldots, N\}$
4: **for** $l = 1$ to $L$ **do**
5:     $\mathrm{prod} \leftarrow \mathrm{prefix\_dp}[1 : N, l - 1] \times \mathrm{matching}[l - 1, 1 : N]$
6:     $\mathrm{prefix\_dp}[1 : N + 1, l] \leftarrow \mathrm{cumsum}(\mathrm{prod}, \mathrm{axis} = 0)$
7: Initialize $\mathrm{suffix\_dp} \leftarrow \mathbf{1}^{(N+1) \times (L+1)}$
8: **for** $l = L - 1$ down to $0$ **do**
9:     $\mathrm{prod} \leftarrow \mathrm{suffix\_dp}[1 : N + 1, l + 1] \times \mathrm{matching}[l, 1 : N]$
10:    $\mathrm{suffix\_dp}[1 : N, l] \leftarrow \mathrm{cumsum}(\mathrm{prod}, \mathrm{axis} = 0)$
11: $\mathrm{alignments}[l] \leftarrow \sum_{i=1}^{N} \mathrm{prefix\_dp}[i, l] \times \mathrm{suffix\_dp}[i + 1, l + 1]$ for all $l$
12: **return** alignments

---

## I  EXPERIMENT WITH TOY SYNTHETIC DATASET

There is a distribution shift between the sequences SCISOR sees during training - composed of unnatural sequences with potentially many random insertions - and the inputs we wish to use when shrinking natural proteins. To show that this distribution shift is not a fundamental issue for SCISOR, we perform an experiment with a toy synthetic dataset designed to capture this challenge.

Specifically, our toy synthetic data has an alphabet of A, B, C, but assumes that the only functional sequences are those that alternate A and B, such as "ABABABABA". The length of functional sequences is sampled uniformly at random between 1 and 20.

During training, due to the random insertions, SCISOR is exposed mostly to sequences that are not valid, such as "ACBBABCCBAB". Nevertheless, when we apply SCISOR to shrink a valid sequence, it should learn to only return valid subsequences. Indeed, in Fig. 14, we see that this is the case: within half an hour of training, SCISOR samples >99% functional subsequences from functional sequences. All training details are the same as for our protein experiments, except we use a smaller batch size of 64.

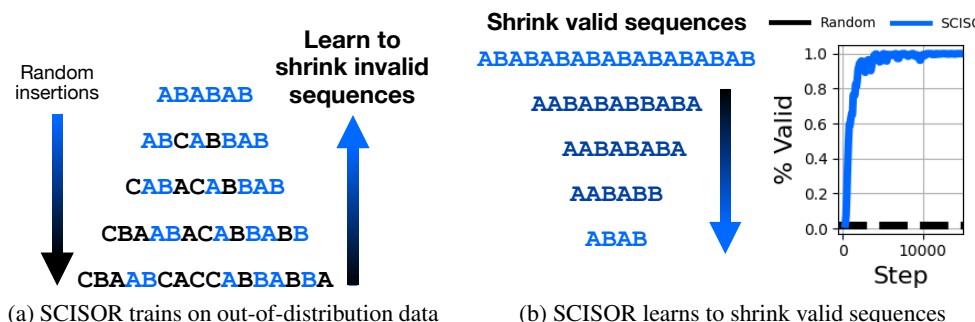

(a) SCISOR trains on out-of-distribution data        (b) SCISOR learns to shrink valid sequences

Figure 14: **Despite being trained on unnatural sequences, SCISOR is able to shrink natural sequences while preserving validity.** (a) We train SCISOR S on a toy synthetic dataset where natural sequences are alternating As and Bs, such that the random insertions noising process results in primarily invalid sequences. (b) We find that despite being trained to shrink invalid sequences, SCISOR quickly learns to shrink valid sequences as well in less than 30 minutes of training. Here, we plot the percent of alternating sequences of length $L = 20$ which SCISOR shrinks to another valid sequence after $M = 10$ deletions.

