# OpenReview forum: "A Diffusion Model to Shrink Proteins While Maintaining Their Function"
_ICLR.cc/2026/Conference — ICLR 2026 Poster_

### Official Review · Reviewer_gx5L · 2025-11-01

**Soundness:** 3
**Presentation:** 3
**Contribution:** 3
**Rating:** 6
**Confidence:** 4

**Summary:**

The paper proposes SCISOR, a discrete diffusion model for shortening proteins by learning to delete residues. The forward process is insertion-only; the reverse denoiser learns to plan deletions that yield sequences resembling natural proteins. A dynamic-programming alignment counting algorithm marginalizes over insertion paths and produces a tractable training objective. Empirically, SCISOR is a competitive protein generative model, achieves SOTA deletion effect prediction on ProteinGym, and, when shrinking real proteins, better preserves predicted foldability and functional motifs than strong baselines. Code and additional analyses are provided.

**Strengths:**

1. Clear, principled formulation of deletion-learning diffusion. The insertion-only forward / deletion reverse construction is elegant and purpose-built for length reduction, and it is distinct from prior substitution-only diffusion models in sequence generation.
2. Strong deletion-specific evidence. SCISOR reaches SOTA on ProteinGym deletion effect prediction and shows consistent wins in structure/motif preservation during shrinking.
4. Useful case study and qualitative analyses. The RalA example and the comparison to deletions observed in natural alignments help interpret what the model is deleting.
5. Implementation details, ablations, and code release are documented.

**Weaknesses:**

1. **Practical claim (“preserve function while shrinking”) is only partially supported.**
Evidence for function preservation relies mainly on (i) ProteinGym deletion assays, which typically remove very few sites per sequence, and (ii) structure-centric proxies (pLDDT/TM/RMSD) for larger shrinkage experiments. These are helpful signals but do not fully validate functional retention at scale. The paper does measure functional-site enrichment, which is a step in the right direction, but this is confined to annotated motifs and could be expanded. Overall, the story is stronger on “planning deletions” than on “confirmed function after large deletions.”

2. **Evaluation granularity for shrinking could be deepened.**
For the 200-sequence study, instead of global pLDDT/TM/RMSD curves, more  function-proximal evidence could also be considered in the main paper: e.g., family-stratified analysis of active site/binding pocket conservation, ligand-contact residue retention, or catalytic triad/metal-coordination integrity.

3. **Distribution shift between training noise and inference use-case.**
The model is trained to reverse insertions into natural sequences, whereas shrinking deletes from a real protein directly. While the authors acknowledge limitations and future directions; some empirical evidence on the robustness would be helpful (e.g., proteins with long disordered tails vs compact cores)

4. **Compute at inference scales with deletion count.**
The paper discusses efficient sampling, but more wall-clock comparisons against strong non-diffusion baselines (e.g., ProGen2-based pruning strategies) on long proteins would clarify practicality.

**Questions:**

1. Would it be possible to include residue-level functional analyses for larger deletions—for example, checking whether annotated catalytic, binding, or interface residues are retained after shrinking, and whether their local structural geometry is preserved?
2. Are there examples where the model maintains good global pLDDT/TM-score but loses key functional residues or pocket geometry? Showing a few such cases would help clarify the limits of the method and when shrinking becomes unsafe.
3. The number of inference steps scales with the number of planned deletions. When would this becomes an issue in practice, i.e., what's the computational cost comparison of different methods in terms of different protein length and deletion length?

---

> ### Author Response · Authors · 2025-11-22
> **Rebuttal (1/2)**
>
> Thank you for your thoughtful review! Below we address your concerns regarding the soundness of the training procedure, and the choice of validation and baselines, and answer your questions. With regards to framing, we have tried to make our claims as specific and tempered as possible in the paper; we would be happy to change any specific phrases you find to the contrary.
>
> > W1. Practical claim (“preserve function while shrinking”) is only partially supported. Evidence for function preservation relies mainly on (i) ProteinGym deletion assays, which typically remove very few sites per sequence, and (ii) structure-centric proxies (pLDDT/TM/RMSD) for larger shrinkage experiments. These are helpful signals but do not fully validate functional retention at scale. The paper does measure functional-site enrichment, which is a step in the right direction, but this is confined to annotated motifs and could be expanded. Overall, the story is stronger on “planning deletions” than on “confirmed function after large deletions.”
>
> > W2. Evaluation granularity for shrinking could be deepened. For the 200-sequence study, instead of global pLDDT/TM/RMSD curves, more function-proximal evidence could also be considered in the main paper: e.g., family-stratified analysis of active site/binding pocket conservation, ligand-contact residue retention, or catalytic triad/metal-coordination integrity.
>
> > Q1. Would it be possible to include residue-level functional analyses for larger deletions—for example, checking whether annotated catalytic, binding, or interface residues are retained after shrinking, and whether their local structural geometry is preserved?
>
> While the function-proximal metrics you mention may be more accurate than the global metrics we report, *they are not available for a large variety of protein families*. Our global metrics allow us to compare in silico indicators of function for a broad class of proteins, allowing us to evaluate the *generalizability* of SCISOR and baseline methods.
>
> As you note in your strengths, in App E however we do a deep dive into RalA whose function is sensing GTP. In this case we devised more specific function predictors by measuring the preservation of the binding site. We see similar results in that setting as we did in our large-scale experiments in Sec 7 and 8.
>
> > W3. Distribution shift between training noise and inference use-case. The model is trained to reverse insertions into natural sequences, whereas shrinking deletes from a real protein directly. While the authors acknowledge limitations and future directions; some empirical evidence on the robustness would be helpful (e.g., proteins with long disordered tails vs compact cores)
>
> Please see **Distribution shift** in the global response, where we give both theoretical and empirical evidence that we have overcome the effects of the distribution shift. These include new results.
>
> We have also stratified our results based on structural info in **new results** (App F.4). These show that SCISOR is robustly shrinking proteins regardless of their foldability. Apps D and E are also case studies on compact proteins.
>
> > W4. Compute at inference scales with deletion count. The paper discusses efficient sampling, but more wall-clock comparisons against strong non-diffusion baselines (e.g., ProGen2-based pruning strategies) on long proteins would clarify practicality.
>
> > Q2. The number of inference steps scales with the number of planned deletions. When would this becomes an issue in practice, i.e., what's the computational cost comparison of different methods in terms of different protein length and deletion length?
>
> To shorten a length 200 protein by 100 letters on an NVIDIA-A100 80GB GPU, we note, to the first significant-digit, ProGen2-base takes about 7 seconds, Raygun takes about 7 seconds, and SCISOR-large also takes about 7 seconds. If $L$ is the length of the protein and $M<L$ is the number of deletions, ProGen2-base scales as $O(L)$, Raygun scales as $O(1)$, and SCISOR-large scales as $O(M)$ in function evaluations. All methods are fast enough for practical protein design applications.

---

> > ### Author Response · Authors · 2025-11-22
> > **Rebuttal (2/2)**
> >
> > > Q2. Are there examples where the model maintains good global pLDDT/TM-score but loses key functional residues or pocket geometry? Showing a few such cases would help clarify the limits of the method and when shrinking becomes unsafe.
> >
> > Yes, absolutely. Evidence for this exists in Fig 6, where ProGen2 preserves more functional motifs, but SCISOR more often keeps the structure preserved (note this reflects an idiosyncrasy of the Progen2 baseline’s assumptions rather than a bone fide tradeoff between methods. This is discussed in the response to Xm99.) We can also see in Sec D that at high shrinking percentages, SCISOR prefers to delete one domain of a two-domain protein – while this leads to a natural-looking shrunk sequence, it likely destroys function.
> >
> > These are known limitations of using evolutionary information to guide design. Yet these sorts of methods are extremely powerful and popular in the common setting that we do not have much information about a protein of interest. Given structural or experimental information about function, a protein design expert would combine them with evolutionary information using methods like BoltzGen or LaMBO2. One can do the same with SCISOR using guidance.

---

### Official Review · Reviewer_xiN6 · 2025-11-01

**Soundness:** 2
**Presentation:** 2
**Contribution:** 3
**Rating:** 4
**Confidence:** 3

**Summary:**

This paper introduces SCISOR, a generative diffusion model designed for protein shrinking, learning to remove amino acids from natural proteins while maintaining functional and structural integrity. The proposed model reformulates sequence diffusion where the forward process inserts tokens, and the reverse process learns to delete them. The authors derive an exact training objective with two KL terms and compute analytical expressions for event intensity.The paper also demonstrates competitive performance on protein sequence modeling (UniRef50/90), strong correlation with natural deletion patterns, and superior structural preservation in protein shrinking tasks.

**Strengths:**

1. The paper extends diffusion models to discrete, length-varying domains, a major conceptual leap beyond standard fixed-length sequence models. The application to protein shrinking which is previously dominated by self-supervised or autoregressive models is novel and impactful.
2. The theoretical foundation is rigorous, derivations are mathematically sound. Experimental validation is multifaceted, spanning sequence likelihoods, structure preservation, and biological realism.

**Weaknesses:**

1. Basically, this work simply apply masked diffusion to protine domain. And the loss function designed is similar to the loss designed in [1]. It seems this work only extend the loss to the protein shrinking setting (as well as Theorem 4.2). This work should further elaborate what new insights does this paper convey.
2. Effects of major components (e.g., π distribution, alignment weighting) is not sysytematically isolated. For example, if π is selected from other priors with a distribution shift, will there be a negative influence? By comparison with a unifrom weighting, is there evidence that the event intensity weighting improves stability or convergence?
3. Section 4.2 acknowledges a distribution shift between training on noisy inserted sequences and sampling deletions from realistic proteins. Is there a way to conduct quantitative analysis of this mismatch?
4. Hyperparameters are briefly mentioned, but full training details (compute hours, dataset splits) are incomplete.

[1] Why masking diffusion works: Condition on the jump schedule for improved discrete diffusion.

**Questions:**

Refer to Weaknesses

---

> ### Author Response · Authors · 2025-11-22
> **Rebuttal**
>
> Thank you for your thoughtful review! Below we address your concerns regarding the novelty of our method and provide new ablations.
>
> > W1. Basically, this work simply apply masked diffusion to protine domain. And the loss function designed is similar to the loss designed in [1]. It seems this work only extend the loss to the protein shrinking setting (as well as Theorem 4.2). This work should further elaborate what new insights does this paper convey.
>
> Our theoretical and engineering contributions (outlined in Sec 3) allow us to build one of the first large-scale variable length sequence diffusion models in any setting. The challenges that previous practitioners faced were theoretical – such as how to build a diffusion model when there is no stationary distribution (Thm 4.1), and deriving the loss (Thm 4.2) – as well as practical – we also had to Rao-Blackwellize the gradient estimator, something that was overlooked in TDDM for example; this step was mandatory to make the model train efficiently as we show below. Crudely, it is clear that our contributions are highly non-trivial given the 5 pages of proofs and algorithms in just Apps G and H.
>
> Even our results on proteins inform the design and use of diffusion models more broadly. Before these results, diffusion models were popular largely for their fast generation in language, and for their utility in conditional generation. Our results show that diffusion models can have inductive biases that allow them to perform state-of-the-art prediction on downstream tasks as well (Sec 7, 8).
>
> > W2. Effects of major components (e.g., π distribution, alignment weighting) is not sysytematically isolated. For example, if π is selected from other priors with a distribution shift, will there be a negative influence? By comparison with a unifrom weighting, is there evidence that the event intensity weighting improves stability or convergence?
>
> In **new results**, in App. F.5 we ablate $\pi$ and our Rao-Blackwellization (“event intensity weighting”). We see that without our Rao-Blackwellization, the model performs near trivially, showing that the Rao-Blackwellized training objective is a critical piece to making this model work in practice. $\pi$ on the other hand made a minor but noticeable difference.
>
> > W3. Section 4.2 acknowledges a distribution shift between training on noisy inserted sequences and sampling deletions from realistic proteins. Is there a way to conduct quantitative analysis of this mismatch?
>
> Please see **Distribution shift** in the global response, where we give both theoretical and empirical evidence that we have overcome the effects of the distribution shift. These include new results.
>
> > Hyperparameters are briefly mentioned, but full training details (compute hours, dataset splits) are incomplete.
>
> In App C.3 we describe the dataset splits on Uniref 50 and 90, and state “The SCISOR S and M models were trained for about one week each on two NVIDIA A100 GPUs. The SCISOR L model was trained for about four days on four NVIDIA H100 GPUs”.

---

### Official Review · Reviewer_RVrd · 2025-11-05

**Soundness:** 3
**Presentation:** 3
**Contribution:** 3
**Rating:** 4
**Confidence:** 4

**Summary:**

The paper proposes SCISOR, a discrete diffusion model that learns to shorten protein sequences while preserving their structural and functional integrity. Instead of using additive noise like standard diffusion models, SCISOR defines insertions as the forward process and deletions as the reverse process, allowing it to model sequence shrinking directly.  Theoretically, the author prove that a stationary distribution is unnecessary for defining a valid diffusion process, derive a schedule-conditioned loss for stable training and introduce a Rao-Blackwellized gradient estimator by integrating over all insertion paths via sequence alignment. Empirically, SCISOR achieves competitive perplexity and structure-based scores compared to leading generative baselines, and demonstrates superior preservation of foldability and active sties when applied to protein shortening tasks. Overall, the work establishes a principled and scalable deletion-based diffusion framework for biological sequence generation

**Strengths:**

The paper is original in both formulation and problem scope. It extends discrete diffusion to irreversible deletion processes by defining a continuous-time insertion-only forward process and a deletion-based reverse process, effectively removing the symmetry assumption of prior discrete diffusion models. This enables a principled treatment of the protein shortening problem, a biologically relevant task that has not been previously modeled in this way. The theoretical development is coherent and well integrated, combining the continuous-time formulation, schedule-conditioned objective, and alignment-based gradient estimator into a unified framework. Empirically, the model achieves good results on both sequence and structure-based evaluations. The paper is clearly written, effectively linking theoretical innovation with practical biological relevance.

**Weaknesses:**

Despite the paper’s strong theoretical formulation, its empirical evidence for true functional preservation remains incomplete. The following issues address key gaps between the model’s assumptions, its training evaluation setup, and its stated goal of maintaining biological function after large-scale deletions.

1. Validation Gap
**Limited experimental validation**: ProteinGym provides experimental measurements only for 1-3 deletions, but the main application involves 20-50% shrinking (50-100+ residues). While computational metrics show SCISOR outperforms baselines at large scales (Fig. 6), there's no experimental validation that these predicted improvements on large-scale deletions correlate with actual functional retention.
**Unvalidated surrogate metrics**: For large-scale shrinking, the paper relies entirely on predicted metrics without validating their correlation with actual function. Additionally, structure predictions use suboptimal parameters: OmegaFold with 1 cycle instead of recommended 10 cycles, acknowledged to produce "lower overall pLDDT scores" (App. C.4.3). No validation that method rankings remain consistent with proper parameters.

2. Distribution shift underexplored
 Training uses sequences with random insertions while inference uses natural proteins. Authors acknowledge this (Sec. 5) but provide only one validation example (R4SNK4, App. D) showing modest correlations (Fig. 7b, mostly <0.5). More examples across diverse protein families would strengthen the claim that the model learns meaningful evolutionary patterns despite this mismatch.

3. Fundamental Gap
**"Natural-looking" Does Not Guarantee Functional Preservation**:   SCISOR optimizes for shrunk sequences that look "natural" (high q_θ(X̃)), but this does not guarantee they maintain the original protein's function. A shrunk sequence can achieve high naturalness by resembling a common domain found in nature, while completely losing other functional domains of the original protein. Example limitation: Consider a 500-residue transcription factor with a DNA-binding domain (100 aa) and activation domain (400 aa). SCISOR might produce a 200-residue shrunk version containing only the DNA-binding domain. This would score high on naturalness and structural metrics (pLDDT, TM), but would completely lose transcriptional activation function.
**Metrics don't validate functional retention** (1) pLDDT measures foldability, not specific function. (2) TM score measures structural similarity, but high TM doesn't guarantee functional preservation, e.g., deleting an allosteric regulatory domain can maintain high TM while abolishing regulation. (3) "Functional enrichment" only checks if annotated sites exist, not whether they remain functional in the altered structural context.
**Acknowledged but understated**: The paper admits "similar sequences are likely to have the same function, but this is not guaranteed" (Sec. 9), yet frames SCISOR as a protein engineering tool throughout. This gap between "generating natural-looking subsequences" and "preserving specific function" is fundamental to the method's practical utility but receives insufficient discussion.

**Questions:**

Considering the weaknesses outlined above, I have a few questions and suggestions that could help clarify the empirical evidence and strengthen the paper’s claims if addressed in the rebuttal.

1. Do you have any evidence that large-scale (20–50%) deletions predicted by SCISOR retain actual biochemical function?
2. Have you checked whether OmegaFold results and model rankings remain consistent when using the optimal settings ( such as 10 cycles instead of 1)?
3. How robust is SCISOR to the mismatch between insertion-noised training sequences and natural inference inputs?
4. Can you provide validation on more proteins or families beyond the single R4SNK4 example?
5. Since SCISOR optimizes for natural-looking sequences, how do you ensure that key functional domains are not deleted?

---

> ### Author Response · Authors · 2025-11-22
> **Rebuttal (1/2)**
>
> Thank you for your thoughtful review! Below we address your concerns regarding the soundness of the training procedure, and the choice of validation, and answer your questions. With regards to framing, we have tried to make our claims as specific and tempered as possible in the paper; we would be enthusiastic to change any specific phrases you find to the contrary.
>
> > W1a. Limited experimental validation: ProteinGym provides experimental measurements only for 1-3 deletions, but the main application involves 20-50% shrinking (50-100+ residues). While computational metrics show SCISOR outperforms baselines at large scales (Fig. 6), there's no experimental validation that these predicted improvements on large-scale deletions correlate with actual functional retention.
>
> > Q1. Do you have any evidence that large-scale (20–50%) deletions predicted by SCISOR retain actual biochemical function?
>
> We (1) clarify our claims, (2) clarify that the small deletion results on ProteinGym are significant, (3) clarify that our in-silico results say are informative for large shrinking, (4) discuss the prospects of wet lab experiments, and (5) discuss evidence that proteins can be shrunk by 20-50%.
>
> 1. We do not claim that every or even most proteins can be shrunk by 20-50%. The question of how much any individual protein can be shrunk is an interesting question for future research whose *inquiry is enabled by SCISOR*: while there is ample evidence that all our baselines produce non-functional proteins with even a small number of deletions, SCISOR is able to produce plausible designs that are much smaller, enabling experimentalists to efficiently search the space of shrunk sequences in the future.
>
> 2. Our state-of-the-art results for small numbers of deletions in the lab is a significant and main contribution in its own right. Predicting the effects of single deletions accurately can be used to interpret clinical variants or understand the functional parts of a protein sequence. This is indeed why the ProteinGym benchmark was developed.
>
> 3. Furthermore, the in-silico gap between SCISOR and other methods is so large we believe it is unlikely that they will not manifest in improved designs in practice. In Fig 6 we show that all methods perform roughly as well as random, while SCISOR shows hardly any harm to pLDDT or RMSD up to 20% shrinkage across protein families. We also note that these in-silico metrics were the breadth of analysis in a number of impactful machine learning papers, such as EvoDiff, ProGen, DPLM, and many more.
>
> 4. As well, please see **wet lab experiments** in the global response, where we describe the challenges of building an appropriate experimental assay, and recap the potential impact of our model in the absence of those results.
>
> 5. Finally, it is trivial to find proteins, such as very long transcription factors, which are known to function with >50% deletion in their disordered region. Showing we can shrink a single protein by 50% is not necessarily meaningful therefore.
>
> > W1b. Unvalidated surrogate metrics: For large-scale shrinking, the paper relies entirely on predicted metrics without validating their correlation with actual function.
>
> pLDDT, TM, and RMSD are standard in-silico metrics and their relationship with function is thoroughly explored and validated in previous literature, [example](https://doi.org/10.1093/bioinformatics/btq066). In particular, they are the typical benchmarks used ubiquitously for validation in impactful machine learning methods such as EvoDiff, ProGen, DPLM, RayGun and many more. In App E we also look at more granular metrics such as DockQ, another thoroughly explored metric. Our other metric was functional retention, using annotated functional or binding sites. Indeed these metrics have also been thoroughly [explored in the literature](https://doi.org/10.1093/nar/gkaa913).
>
>
> > Q2. Have you checked whether OmegaFold results and model rankings remain consistent when using the optimal settings ( such as 10 cycles instead of 1)?
>
> We did Omegafold with 1 loop because of the scale of our experiments; we observed that increasing the number of loops simply increased the pLDDT across structures. In a **new experiment** we compared pLDDTs for 200 wild type and shrunk designs from 1 vs 10 cycles. The Pearson correlation was 0.95, with a slope of 0.93 and intercept of 8.3. Therefore simply adding 8.3 to all pLDDT numbers is an accurate way to approximate our results had we run them with more cycles. We therefore expect that repeating our experiments with 10 cycles would not affect our conclusions.
>
> In smaller scale experiments on RalA, we did use Alphafold which provides a more expensive yet robust structural prediction. We found similar results in this case as well.

---

> > ### Author Response · Authors · 2025-11-22
> > **Rebuttal (2/2)**
> >
> > > W2: Distribution shift underexplored: Training uses sequences with random insertions while inference uses natural proteins. Authors acknowledge this (Sec. 5) but provide only one validation example (R4SNK4, App. D) showing modest correlations (Fig. 7b, mostly <0.5). More examples across diverse protein families would strengthen the claim that the model learns meaningful evolutionary patterns despite this mismatch.
> >
> > > Q3: How robust is SCISOR to the mismatch between insertion-noised training sequences and natural inference inputs?
> >
> > > Q4: Can you provide validation on more proteins or families beyond the single R4SNK4 example?
> >
> > Please see **Distribution shift** in the global response, especially paragraph (3), where we give both theoretical and empirical evidence that we have overcome the effects of the distribution shift. These include new results on two other qualitatively different protein examples beyond R4SNK4 and the significance of our findings (our correlations have a p-value of <1e-10).
> >
> > > W3a. "Natural-looking" Does Not Guarantee Functional Preservation: SCISOR optimizes for shrunk sequences that look "natural" (high q_θ(X̃)), but this does not guarantee they maintain the original protein's function.
> >
> > > W3c. Acknowledged but understated: The paper admits "similar sequences are likely to have the same function, but this is not guaranteed" (Sec. 9), yet frames SCISOR as a protein engineering tool throughout.
> >
> > > Q5 Since SCISOR optimizes for natural-looking sequences, how do you ensure that key functional domains are not deleted?
> >
> > This is not a weakness but a strength. We explain that (1) the insight that naturalness can be used for design has been a boon for protein engineering, (2) other functional information, when available, can be used in a complementary way to evolutionary information, and (3) our experiments indeed show that SCISOR likely preserves function in practice.
> >
> > 1. While naturalness is not identical to function, it is a staple in modern protein engineering because we often have little other information about the function of a protein. The same criticism can be levied against ESM, EVE, and to models as far back as EVCouplings; the reason these models have been hugely successful is because naturalness *correlates* with function. SCISOR extends this ubiquitous tool used throughout protein engineering to shrinking. The theoretical mismatch you mention is therefore well known and litigated. Methods that only use conservation are used to [build new proteins](https://www.science.org/doi/10.1126/science.aba3304), [optimize antibodies](https://www.nature.com/articles/s41587-023-01763-2), [identify pathogenic variants](https://www.nature.com/articles/s41586-021-04043-8), and much more. These methods have also been used to design proteins with very large mutations: [30% difference to wild-types](https://www.science.org/doi/10.1126/science.aba3304) up to [almost 70%](https://www.science.org/doi/10.1126/science.ads0018).
> >
> > 2. When more information about function is available, such as structure or experiments, they are combined with evolutionary models (ex. BoltzGen or LaMBO1/2). Incorporating other functional information is therefore established as a complementary task, and it can be done with SCISOR using diffusion guidance for example.
> >
> > 3. Our experimental results also look at how often function is preserved during shrinking. Our ProteinGym results, preservation of functional motifs (Fig 6d), and deep dive into RalA (App E), all show that naturalness is a useful design criterion for the task of shrinking, in particular leading to state-of-the-art designs, with a large gap to baselines.
> >
> > > W3b. Metrics don't validate functional retention (1) pLDDT measures foldability, not specific function. (2) TM score measures structural similarity, but high TM doesn't guarantee functional preservation, e.g., deleting an allosteric regulatory domain can maintain high TM while abolishing regulation. (3) "Functional enrichment" only checks if annotated sites exist, not whether they remain functional in the altered structural context.
> >
> > These are general metrics that correlate with function preservation across protein families; a more specific function preservation experiment at large scale is currently impossible. This is because for almost every protein, there’s no good way to predict its function in silico, so we have to rely on the best metrics we have if we want to test across protein families. Moreover, our state-of-the-art performance on deletion effect prediction in ProteinGym confirms that SCISOR’s proposed deletions best preserve function (as measured in experimental assays) for 1-3 deletions.
> >
> > In App E, we do a deep dive into RalA whose function is sensing GTP. In this case we devised more specific function predictors by measuring the preservation of the binding site. We see similar results in that setting as we did in our large-scale experiments in Sec 7 and 8.

---

### Official Review · Reviewer_Xm99 · 2025-11-05

**Soundness:** 4
**Presentation:** 4
**Contribution:** 3
**Rating:** 8
**Confidence:** 4

**Summary:**

SCISOR is a discrete diffusion framework for protein miniaturization. The forward process inserts residues into natural sequences; the learned reverse process deletes them, enabling principled exploration of the deletion space. Trained on large protein corpora with an ESM-style backbone, SCISOR attains strong accuracy on deletion-effect prediction (ProteinGym) and, for design, shrinks diverse UniProt proteins while preserving in-silico foldability (pLDDT), structural similarity (TM/RMSD), and functional-site retention better than LM baselines at moderate shrinkage.

**Strengths:**

- Right inductive bias for deletions. By making the forward process insertions, the reverse process naturally learns deletions. That’s a clean, principled way to search the combinatorial deletion space.

- Design-oriented evaluation. The UniProt shrinking study checks multiple in-silico proxies (pLDDT, TM/RMSD, motif enrichment) and shows consistent gains over ProGen2 and Raygun for realistic shrink amounts.

- Scalable implementation details. Uses ESM2 + flash attention; training/data choices are clear enough to be reproducible.

- Clarity about scope. The paper is explicit that SCISOR currently performs deletions only, which makes contributions and limitations crisp.

**Weaknesses:**

- Wet-lab validation is limited. The main evidence for design quality is in-silico (structure confidence/similarity, motif enrichment). Experimental assays beyond ProteinGym measurements would strengthen claims (e.g., stability, activity, expression yields).

- Function vs. foldability tradeoffs. At very aggressive shrinkage (e.g., ~50%), some baselines can retain more annotated sites while losing foldability; SCISOR’s behavior under extreme compression could be characterized more deeply (kinetics, dynamics, allostery).

- Backbone dependence. Performance and generalization lean on the ESM2 backbone and training corpus; robustness across families with sparse data or disordered regions isn’t fully explored.

- Template-free but prior-bound. Although not template-conditioned, SCISOR is still guided by evolutionary priors; novel functions that deviate from natural sequence statistics may remain hard.

**Questions:**

- Generalization tests: How does performance vary across enzymes, membrane proteins, or multi-chain interfaces, where small deletions can disrupt dynamics or binding? Any oligomeric/complex benchmarks planned?

- Experimental readouts: Can you report ΔTm, expression yields, solubility, etc. for a set of shrunk designs to calibrate pLDDT/TM against real biophysics?

- Search strategy: For multi-deletion design, how sensitive are results to sampling vs. greedy selection? Any gains from guidance (classifier-free or property predictors) to target function preservation explicitly?

- Boundaries of shrinkage: Where is the failure frontier (e.g., % deletion where foldability or motif retention collapses) across diverse families?

---

> ### Author Response · Authors · 2025-11-22
> **Rebuttal (1/2)**
>
> Thank you for your thoughtful review! Below we address your concerns regarding the robustness of SCISOR, and the choice of validation, and answer your questions.
>
> > W1. Wet-lab validation is limited. The main evidence for design quality is in-silico (structure confidence/similarity, motif enrichment). Experimental assays beyond ProteinGym measurements would strengthen claims (e.g., stability, activity, expression yields).
>
> Please see **wet lab experiments** in the global response, where we describe the challenges of building an appropriate experimental assay, and recap the potential impact of our model in the absence of those results.
>
> > W2. Function vs. foldability tradeoffs. At very aggressive shrinkage (e.g., ~50%), some baselines can retain more annotated sites while losing foldability; SCISOR’s behavior under extreme compression could be characterized more deeply (kinetics, dynamics, allostery).
>
> That the ProGen2 baseline retains more motifs than SCISOR reflects an idiosyncrasy of its assumptions rather than a bone fide tradeoff between methods. This is reflected in the deep characterization of RalA in App E.
>
> The ProGen2 baseline makes the unrealistic assumption that the effects of deletions are independent. Therefore it artificially inflates the preservation of motifs that it predict are important in the starting sequence even if they are non-functional in the shrunken sequence. Indeed when we implement the ProGen2 baseline re-doing predictions after each deletion, it no longer preserves motifs as well (App F.3).
>
> In App E we evaluate the predicted binding of shrunken RalA to GTP, requiring *both* the preservation of functional motifs and a fold-able protein. Indeed SCISOR builds shrunken protein that are much more often predicted to be functional than all baselines.
>
> > W3. Backbone dependence. Performance and generalization lean on the ESM2 backbone and training corpus; robustness across families with sparse data or disordered regions isn’t fully explored.
>
> > Q1. Generalization tests: How does performance vary across enzymes, membrane proteins, or multi-chain interfaces, where small deletions can disrupt dynamics or binding? Any oligomeric/complex benchmarks planned?
>
> We investigate dependence in (1) App F.1, which shows ProteinGym results stratified by function, and sparsity of data measured by MSA depth, (2) **new results** in App F.6 we show shrinking results stratified by predicted disorder (pLDDT of wt sequence) and data sparsity / ESM2 bias; these results show that our results in Sec. 7 look similar across all conditions, and (3) new results in App F.6 which look at correlations between cellular location and shrinking performance. Finally, SCISOR is currently built for the analysis of single chains, but we anticipate with some fine-tuning on sets of sequences from complexes, it could also shrink complexes.
>
> For (2), we measured data sparsity and EM2 bias using the ESM2 pseudo-likelihood on the starting sequence. We did so inspired by the recent paper characterizing ESM2’s bias using this statistic (https://openreview.net/pdf?id=UvPdpa4LuV), which correlates with the abundance of each protein in the ESM2 training data.
>
> For (3), we investigated your hypothesis that membrane proteins are harder to shrink. In App F.6 we do not see this in our metrics; this may be because many membrane proteins have large extra-membrane regions. While investigating features that were enriched in sequences that were harder to shrink, we found that extracellular proteins were slightly enriched. Future work may investigate this further.
>
> > W4. Template-free but prior-bound. Although not template-conditioned, SCISOR is still guided by evolutionary priors; novel functions that deviate from natural sequence statistics may remain hard.
>
> Sequence design throughout synthetic biology is guided by evolutionary data, and SCISOR extends this prior information to guide deletions. Guiding design using other information, such as structure (e.g., using folding models such as BoltzGen), or experimental measurements (using BayesOpt methods such as LaMBO2) have been established as complementary problems. These later methods are mandatory to specify novel functions.

---

> > ### Author Response · Authors · 2025-11-22
> > **Rebuttal (2/2)**
> >
> > > Q2. Experimental readouts: Can you report ΔTm, expression yields, solubility, etc. for a set of shrunk designs to calibrate pLDDT/TM against real biophysics?
> >
> > We expect the correlation to strongly depend on the particular protein of interest. There is a large literature exploring the connections between these metrics and biophysical quantities. For instance, [this paper](https://journals.plos.org/plosone/article?id=10.1371/journal.pone.0282689) shows that for a particular protein, a reduction in pLDDT by 10 on average results in a reduction of the folding free energy $\Delta\Delta G$ by 1.7 kcal/mol; [this paper](https://arxiv.org/pdf/0705.1490) classifies a mutation that causes a change of more than 0.5 kcal/mol “destabilizing”, we can conclude that even a few points drop in pLDDT could harm a protein’s function. Taking these numbers for granted, Fig 7a suggests that SCISOR could shrink proteins up to 20% while other methods may struggle with 1-3%; this range matches the results of our App E as well.
> >
> > > Q3. Search strategy: For multi-deletion design, how sensitive are results to sampling vs. greedy selection? Any gains from guidance (classifier-free or property predictors) to target function preservation explicitly?
> >
> > We add a **new result** comparing SCISOR to a greedy version of SCISOR on ProteinGym, finding that SCISOR’s sampling and ability to plan help it make better predictions. We implemented a “greedy” SCISOR baseline by simply setting $M=1$ so that SCISOR greedily chooses deletions without planning ahead. This resulted in a drop of spearman on multi-deletion from 0.520 to 0.508. This shows that SCISOR effectively predicts epistasis and plans its deletions. Our greedy results for the shrinking benchmarks in Sec 8 are in App F.2.
> >
> > Above we mentioned incorporating structure or experimental information. Guidance is a classical method to incorporate this data, as shown in LaMBO2. Therefore we predict that guidance will help if functional information is available.
> >
> > > Q4: Boundaries of shrinkage: Where is the failure frontier (e.g., % deletion where foldability or motif retention collapses) across diverse families?
> >
> > This depends on the particular functional metric one uses. For nearly every protein, there is no accurate way to predict whether their function is preserved, making a large-scale analysis beyond our analysis in Sec 7 and 8 not very informative or potentially misleading. In Sec E we provide a deep dive for a variety of metrics for RalA, showing evidence of potentially 10-20% shrinkage.

---

### Official Review · Reviewer_Ryfk · 2025-11-10

**Soundness:** 3
**Presentation:** 3
**Contribution:** 3
**Rating:** 4
**Confidence:** 3

**Summary:**

This paper introduces SCISOR, a new discrete diffusion framework designed to “shrink” protein sequences by learning to delete amino acids while maintaining functional integrity. The core idea is to train a denoiser to reverse a pure insertion-only forward process (a pure birth process), effectively teaching the model how to delete. SCISOR is trained on evolutionary sequence data, learning to remove randomly inserted amino acids to recover natural sequences. The trained denoiser is then applied to shorten real protein sequences.

The authors claim that SCISOR achieves state-of-the-art deletion-effect prediction on ProteinGym, and that its suggested deletions preserve functional motifs and structural integrity better than prior generative sequence models such as ProGen2 and Raygun.

**Strengths:**

1. **Motivation relevance**: The paper tackles a meaningful biotechnological problem—designing shorter, more manufacturable proteins.
2. **Technical creativity**: The proposed insertion-only diffusion and the use of sequence alignment to marginalize over insertion paths is elegant and mathematically nontrivial.
3. **Empirical performance**: SCISOR performs well on ProteinGym deletion benchmarks, outperforming established baselines in both single and multi-deletion effect prediction.

**Weaknesses:**

1. **Training–application mismatch**:The model is trained to reverse random insertions ($p(X_0|X_0+noise)$) but applied to delete residues from real sequences ($p(X_{shrunken}|X_0)$).This mismatch creates a serious distribution shift, acknowledged but not resolved. It remains unproven that a denoiser trained to remove random noise can effectively identify biologically nonessential regions for deletion.
2. **Unfair or weak baselines**: The main “shrinking” baseline (ProGen2) assumes independent single deletions—a simplified and unrealistic setup. Although a more accurate sequential baseline ($O(L⋅M)$) is mentioned in Appendix F.3, it is not the focus of comparison, and key results (e.g., TM, RMSD, Fnc Enrich) for that stronger baseline are missing.
3. **Computational cost**: Training SCISOR is extremely expensive. Computing $p(prev(X_t) | X_0, X_t, M_t)$ requires dynamic programming over all deletion alignments $(O(|X_0||X_t|)$, repeated for each batch. This forced the authors to apply window approximations, potentially biasing the training.
4. **Functional preservation remains indirectly validated**: The assumption that “naturalness” (model likelihood) correlates with function retention is reasonable and common in protein LMs, but the current evidence is largely in silico. Including further structural or biochemical proxies — e.g., energy-based metrics, functional-domain conservation, or experimental validation — would significantly enhance the biological credibility of the work.

**Questions:**

1. Could the authors justify—either theoretically or empirically—why reversing random insertions is a good proxy for learning function-preserving deletions?
2. What would happen if the forward process included deletions (as in TDDM) instead of insertions? Would the model’s generative behavior improve or degrade?
3. How sensitive are the shrinking results (Fig. 6) to the sampling strategy for ProGen2 and to the hyperparameters of the corrector steps?
4. Could the authors provide TM, RMSD, and Fnc Enrich metrics for the stronger baseline discussed in Appendix F.3?

---

> ### Author Response · Authors · 2025-11-22
> **Rebuttal (1/2)**
>
> Thank you for your thoughtful review! Below we address your concerns regarding the soundness of the training procedure, and the choice of validation and baselines, and answer your questions.
>
> > W1. Training–application mismatch
>
> > Q1. Could the authors justify—either theoretically or empirically—why reversing random insertions is a good proxy for learning function-preserving deletions?
>
> Please see **Distribution shift** in the global response, where we give both theoretical and empirical evidence that we have overcome the effects of the distribution shift. These include new results.
>
> > W2. Unfair or weak baselines: The main “shrinking” baseline (ProGen2) assumes independent single deletions—a simplified and unrealistic setup. Although a more accurate sequential baseline () is mentioned in Appendix F.3, it is not the focus of comparison, and key results (e.g., TM, RMSD, Fnc Enrich) for that stronger baseline are missing.
>
> > Q2a. How sensitive are the shrinking results (Fig. 6) to the sampling strategy for ProGen2.
>
> > Q3. Could the authors provide TM, RMSD, and Fnc Enrich metrics for the stronger baseline discussed in Appendix F.3?
> We have added full results (including TM, RMSD, and Fnc Enrich) for the expensive O(LM) ProGen2 baseline in App. F.3. We originally used the O(L) ProGen2 baseline because its computational cost is comparable to SCISOR’s, while the O(LM) baseline’s cost is up to 100x higher (12 mins vs 7 seconds on average to shrink a length-200 protein by 50%). Despite being much more computationally expensive, this baseline actually performs worse than our main O(L) ProGen2 baseline in practice. We suspect this is because ProGen2 makes poor predictions on sequences that have already had several deletions applied, as these are out-of-distribution. As a result, we have chosen to keep the existing O(L) baseline in the main text.
>
> > W3. Computational cost: Training SCISOR is extremely expensive. Computing $p(\text{prev}(X_t)\mid X_t, X_0, M_t)$ requires dynamic programming over all deletion alignments $(O(|X_0||X_t|)$, repeated for each batch. This forced the authors to apply window approximations, potentially biasing the training.
>
> We used a number of tricks to reduce the computational cost in theory and in practice, ultimately allowing us to train a state-of-the-art discrete diffusion model with *less* compute than EvoDiff and DPLM. We estimate that DPLM was trained for at least 4x longer than SCISOR and EvoDiff for even longer than that.
>
> When training, we need to compute $p(\text{prev}(X_t)\mid X_t, X_0, M_t)$ for every deletion! However, we can do so with only a *single alignment* between X_t and X_0. Our algorithm for doing so is one of our contributions, described in App H.  In practice, the alignments computation takes up a fraction of the training time compared to the forwards and backwards passes and can be pre-cached.
>
> With respect to our window method, we note that (1) we window not because of the alignment, but because of the need to make predictions for long sequences, a challenge that would face any model that learns to shrink, (2) the window we use is 2048, larger almost any real protein, so it likely doesn’t affect performance on real tasks, and (3) despite the windowing in principle harming generative modelling accuracy, we still achieve high quality likelihoods and samples in Sec. 6.
>
> > W4. Functional preservation remains indirectly validated: The assumption that “naturalness” (model likelihood) correlates with function retention is reasonable and common in protein LMs, but the current evidence is largely in silico. Including further structural or biochemical proxies — e.g., energy-based metrics, functional-domain conservation, or experimental validation — would significantly enhance the biological credibility of the work.
>
> Please see **wet lab experiments** in the global response, where we describe the challenges of building an appropriate experimental assay, and recap the potential impact of our model in the absence of those results.

---

> > ### Author Response · Authors · 2025-11-22
> > **Rebuttal (2/2)**
> >
> > > Q2a. How sensitive are the shrinking results (Fig. 6) to the hyperparameters of the corrector steps?
> >
> > We did not use corrector steps in the shrinking results. This is because corrector steps can add new letters, so we left them out to keep evaluations fair. We only used them for the unconditional sample quality results.
> >
> > > Q4. What would happen if the forward process included deletions (as in TDDM) instead of insertions? Would the model’s generative behavior improve or degrade?
> >
> > Our hypothesis is that shrinking performance would degrade but generative performance would improve. We also note that such a model at large scale would benefit from our mathematical foundation: switching from insertions to deletions would simply require changing how the alignment is performed in the Rao-Blackwellization step.
> >
> > Intuitively, building a protein by only inserting should be easier since our model only sees small proteins rather than massive ones, saving some computation. However, a model that does both insertions and deletions may split its model capacity to improve predictions of insertions over predictions of deletions, potentially harming deletion quality.

---

> > > ### Comment · Reviewer_Ryfk · 2025-11-27
> > >
> > > Thank you for the detailed rebuttal and the additional baseline experiments. While I appreciate the clarifications, I remain concerned about the conceptual mismatch between the training objective (denoising) and the application (shrinking), which I feel the theoretical arguments do not fully resolve. Additionally, the counter-intuitive underperformance of the stronger sequential baseline makes it difficult to verify the method's superiority against an optimized reference.
> > >
> > > Therefore, I will maintain my score.

---

> > > > ### Author Response · Authors · 2025-11-27
> > > > **Response**
> > > >
> > > > Thank you for your response! We would like to ask for some clarification and add that domain shifts such as ours occur throughout machine learning.
> > > >
> > > > We have made two theoretical and two empirical arguments showing that our model can predict natural patterns of deletion and that these predictions are correlated to function. Could you clarify why these do not resolve the concern and what you would like to see?
> > > >
> > > > Domain shifts from training to testing are ubiquitous in popular models in machine learning and protein engineering. Two salient examples are diffusion models used for optimizing antibodies [(paper)](https://www.biorxiv.org/content/10.1101/2022.07.10.499510v5.full.pdf), and the masking / unmasking process ESM (equivalent to de-noising with masking diffusion) to optimize proteins [(paper)](https://www.biorxiv.org/content/10.1101/2024.07.01.600583v1).
> > > > Even LLMs were pre-trained on internet-scale text, and then used for in-context learning!
> > > > Even though there is a distribution shift, these methods and SCISOR each have extensive experimental evidence of state-of-the-art performance on downstream tasks.
> > > >
> > > > With respect to the baselines, we have compared against the only other model that claims to perform deletion (Raygun). We also *invented* two ways of shrinking with Progen2. These are strong baselines: our new Progen2 even beats Raygun! We gave an explanation of the "counter-intuitive" behavior of the more expensive Progen2 baseline. Could you clarify what more you would like to see?

---

### Author Response · Authors · 2025-11-22
**Global rebuttal (1/2)**

We thank all reviewers for their thoughtful comments - we have made changes to the manuscript and mark the new content in blue. We appreciate that reviewers noted this paper solves multiple fundamental mathematical and engineering challenges to develop the first large-scale diffusion model that can accommodate changes in sequence length in any domain (EditFlow and insertion language models were developed at the same time). We apply this method to the problem of shrinking proteins and achieve state-of-the-art performance on ProteinGym for small numbers of deletions and standard in-silico benchmarks for large deletions, therefore building a tool biologists can use right now in the lab and clinic.

In this global response we address points raised by multiple reviewers: first we discuss the strengths of our methods for validation as compared to lab measurements, then we show theoretical and empirical evidence that we overcome the distribution shift in our training procedure

**Wet lab experiments:**

We are delighted that reviewers appreciated our extensive in-silico experiments and wet-lab ProteinGym results. We agree that *thorough* wet-lab testing would further further strengthen the *impact* of our results, but our *claims* in the paper are carefully written to reflect what we have shown. However, we decided against experimental results in this work as *(1)* popular and accessible experimental platforms are insufficient to increase impact, *(2)* our paper has already shown that algorithmic advances can have a large impact on the quality of designs, which is the most appropriate thesis for this venue, *(3)* early experimental results from a collaborator substantiate our points in (1) and (2). Our in-silico evidence by contrast allowed us to test against >50 different assays across a variety of functions in ProteinGym, the structures of 200 proteins all with different folds, and a deep-dive across multiple metrics on RalA in App E.

1. The smallest validation of a few tens of sequences costs upwards of $10’000, potentially within reach, but it (a) would only test a handful of designs, (b) would only test a single protein, and (c) only look at proteins which are easy to manufacture, that is, already short. As an illustration of the challenge of interpreting such results, Raygun demonstrated that they were able to shrink GFP. However, their designs simply cut the tail, undermining their claim of broader generalizability. Our in-silico experiments, since we could perform them at large scale, gave a different picture of Raygun’s generalizability, demonstrating its close-ness to random design.

    We therefore believe that a small-scale set of assays is not illustrative, and may even be misleading given the idiosyncrasies of any individual target. An impactful and logical evaluation would require many designs across multiple enzyme families, or a deep dive into a single protein of interest, which is far beyond an appropriate experiment in a machine learning paper.

2. We have developed highly non-trivial algorithms and shown that they substantially improve the most popular in silico metrics for design. First, similarly validated machine learning models have been impactful in biology (Evodiff, ProGen, DPLM, etc…). Our results not only provide models that are state-of-the-art at predicting the effects of deletions, they also open the door to the development of similarly effective diffusion models in the future. Given that we are submitting to ICLR, we feel these contributions are of interest to the machine learning community, and that a large experimental campaign outlined in (1) would be inappropriate for this audience.

3. An experimental collaborator attempted to shrink IRED1 by 1 or 3 residues using SCISOR and RayGun showed that the SCISOR designs were functional while the RayGun designs were not. Our claim is that SCISOR is state-of-the-art in practical shrinking design and this is evidence to that claim. However, this experiment better *cautions the interpretation of such results* as described in (1). Indeed, changing the target turned RayGun’s impressive performance on GFP to a failure to make a single deletion in IRED.

---

> ### Author Response · Authors · 2025-11-22
> **Global rebuttal (2/2)**
>
> **Distribution shift:**
>
> As acknowledged in our paper, there is a distribution shift between sequences SCISOR observes in training and the natural sequences we wish to shrink in practice. However, this distribution shift does not affect our claims or the potential impact of SCISOR. In the paper we *(1)* gave a theoretical argument in Sec 5 that the neural network trained with our pipeline should learn to shrink natural proteins, *(2)* showed empirical evidence that it does so in Sec 7 and 8. We have also **now added** *(3)* further evidence that deletions proposed by SCISOR match those found in nature, and *(4)* a demonstration in a toy setting confirming our theory: we  show that a SCISOR model, even with severe distribution shift, can easily learn to shrink out-of-distributions onto a constrained set of sequences.
>
> 1. Our design objective is to find substrings $\tilde X$ of a sequence $X$ that could preserve function. Absent other knowledge about how the protein functions, we use natural-ness to guide our design, that is, we’d like substrings of $X$ weighted by their naturalness: $p(\tilde X\mid X)\propto p(\tilde X)$. This is a standard method for biological design. We showed in Sec. 5 that SCISOR’s objective of learning to undo random insertions is mathematically *equivalent* to this design task.
> 2. The theory makes the assumption of enough compute, data, and model capacity to fit the task; note this is standard in all sorts of ML tasks like masked language modelling and representation learning. In practice we see the theory is verified: Sec 7 and 8 we show that SCISOR is *state-of-the-art* in predicting the effects of small deletions in the lab on ProteinGym (with a gap of 0.06 Spearman to the next best unconditional method) and shrinking proteins while preserving in-silico measures of function (other baselines perform closer to random than to SCISOR). While mitigating distribution shift may be an interesting future direction, it has not stopped SCISOR from qualitatively improving on previous models’ performance on important tasks.
> 3. In App D we showed evidence of (1) directly: the deletions suggested by SCISOR highly correlate with those found in nature. Despite noise, we achieve correlations nearing 50%, which are highly significant with a p-value of <1e-10. We have added two new proteins to these results: a membrane protein and a transcription factor, showing similar results.
> 4. In App I we have added experiments on a synthetic toy dataset to demonstrate the theoretical result in (1). Note our goal is to isolate the statistical effect of the distribution shift in our framework, not build a biologically realistic setting.
>
>     With an alphabet of A, B, C, we assume the only functional sequences are those that alternate A and B: “ABABABABA” for example. We now wish to shrink these proteins while maintaining their function: ABABABABA -> BABA for example. We train a SCISOR model with uniform insertions. This is a much more constrained sequence space than the space of functioning proteins so, for instance, ABABAB, after random insertions can become totally OOD: ACBBABCCBAB. Nevertheless, when we apply SCISOR to a functional sequence, it should only return functional subsequences. Indeed we see this is the case: within half an hour of training, SCISOR samples >99% functional subsequences from functional sequences.

---

### Meta-Review · Area_Chair_XGdZ · 2026-01-07

**Summary:**

The paper proposes a novel diffusion framework for protein shrinking. This is a meaningful step toward protein miniaturization, which is a practically important yet difficult problem. Protein shrinking is easy to generate but hard to verify. Therefore, a core outstanding concern from the reviewers is the effectiveness of the in-silico evaluation: “natural-looking” sequences do not guarantee functional preservation, especially under large deletions. As a result, reviewers request wet-lab validation. The authors argue that sequence naturalness correlates with function, which is reasonable for a paper at ICLR.

Multiple reviewers also remain skeptical of the “denoising as shrinking” paradigm. The authors provide theoretical analysis, toy experiments, and additional protein examples showing correlations with natural deletions, but these concerns are not fully resolved.

Nevertheless, the methodological novelty and strong benchmark results outweigh the outstanding concerns. Therefore, I recommend acceptance.

To avoid understating these limitations, the authors are encouraged to discuss the remaining concerns in the main paper, for example by adding a limitations section after the conclusion.

**Reviewer Concerns:**

Concerns addressed:
1. Search strategy (Xm99)
> The authors add results comparing with greedy version, which performs worse than the sampling version.
2. OmegaFold settings (RVrd)
> The authors show 0.95 correlation between 1 and 10 cycles to justify the setting.
3. Novelty relative to masked diffusion (xiN6)
> The authors emphasize this is not masked diffusion. This concern likely stems from misunderstanding.
4. Inference Speed (gx5L)
> The authors clarify that SCISOR is as fast as baselines during sampling.

Concerns partially addressed:
1. Computational cost (Ryfk)
> The authors state that the computational training cost is lower than EvoDiff and DPLM, but provide no empirical evidence to support this claim.

2. Baseline fairness (Ryfk)
> The authors add TM/RMSD/Fnc metrics for stronger baseline ProGen2, partially addressing the concern.

3. Limited wet-lab validation(Xm99, RVrd)
> The authors cite wet-lab experiments from a collaborator that shrink IRED1 by 1 or 3 residues, which partially address the concern. Referring to the ProteinGym benchmark as wet-lab results appears to be a typo or misunderstanding, as these evaluations are still in-silico.

Outstanding concerns:
1. Distribution shift (Ryfk, RVrd, gx5L)
> The authors provide theory, toy experiments, and additional protein examples showing correlations with natural deletions. However, the concern remains that denoising from randomly lengthened sequences to the original sequence does not necessarily learn to preserve protein function. So the effectiveness of the proposed method under large-scale deletions is not theoretically guaranteed.

2. "Natural-looking" does not guarantee functional preservation (RVrd, Ryfk, gx5L)
> The authors defend that sequence naturalness correlates with function in protein engineering and that in-silico metrics are standard. However, this does not resolve the fundamental issue that optimizing for naturalness can yield functionally broken proteins that look natural. The authors themselves acknowledge "similar sequences are likely to have the same function, this is not guaranteed".

**Reviewer Scores:**

Reviewer Ryfk explicitly maintained their score at 4 after rebuttal.
Reviewers Xm99 and gx5L are likely to maintain their scores at 8 and 6, respectively.
Reviewer xiN6 will likely stay at 4 or rise slightly.
Reviewer RVrd may stay at 4 due to remaining concerns, but could increase the score if persuaded by the added analyses.

Overall, due to the two main outstanding concerns, the scores will remain mixed, making the paper borderline.

---

### Decision · Program_Chairs · 2026-01-26

Accept (Poster)